# A multidisciplinary drought catalogue for southwestern Germany dating back to 1801

Mathilde Erfurt[1,4], Georgios Skiadaresis[2], Erik Tijdeman[3], Veit Blauhut[4], Jürgen Bauhus[2], Rüdiger Glaser[1], Julia Schwarz[2], Willy Tegel[5], Kerstin Stahl[4]

[1]Physical Geography, Faculty of Environment and Natural Resources, University of Freiburg, Freiburg, Germany

[2]Silviculture, Faculty of Environment and Natural Resources, University of Freiburg, Freiburg, Germany
[3]Hydrology and Climatology, Institute of Geography, Heidelberg University, Heidelberg, Germany
[4]Environmental Hydrological Systems, Faculty of Environment and Natural Resources, University of Freiburg, Freiburg, Germany
[5]Forest Growth and Dendroecology, Faculty of Environment and Natural Resources, University of Freiburg, Freiburg, Germany

*Correspondence to*: mathilde.erfurt@geographie.uni-freiburg.de

**Abstract.** Droughts are multidimensional hazards that can lead to substantial environmental and societal impacts. To understand causes and impacts, multiple perspectives need to be considered. Many studies have identified past drought events and investigated drought propagation from meteorological droughts via soil moisture to hydrological droughts and some studies have included the impacts of these different types of drought. However, it is not certain whether the increased frequency and severity of drought events in the past decade is unprecedented in recent history. Therefore, we analyze different droughts and their impacts in a regional context using a multidisciplinary approach. We compile a comprehensive and long-term data set to investigate possible temporal patterns in drought occurrence and place recent drought events into a historical context. We assembled a dataset of drought indices and recorded impacts over the last 218 years in southwestern Germany. Meteorological and river-flow indices were used to assess the natural drought dynamics. In addition, tree-ring data and recorded impacts were utilized to investigate drought events from an ecological and social perspective. Since 1801, 20 extreme droughts were identified as common extreme events when applying the different indices. All events were associated with societal impacts. Our multi-dataset approach provides insights into similarities but also the unique aspects of different drought indices.

# 1 Introduction

Droughts are natural hazards with potentially widespread negative consequences for environment and society. As droughts can affect different domains of the water cycle and water deficits often accumulate only slowly over time, they are commonly considered one of the most complex natural hazards (Wilhite 2000). The complexity of droughts is reflected in the numerous definitions of the hazard, developed by a variety of different disciplinary perspectives and needs for different drought management applications. Frequently these definitions are associated with one of several drought types, e.g., meteorological, hydrological, agricultural, and socioeconomic drought (Wilhite and Glantz 1985). To provide a quantitative assessment of different drought events a wealth of indices exists for each drought type (Zargar et al., 2011; WMO and GWP, 2016). For example, meteorological droughts are often evaluated using the Standardized Precipitation Index (SPI; McKee et al., 1993) at various timescales, hydrological droughts can be described calculating the Standardized Streamflow Index (SSI), agricultural droughts can be assessed using the Soil Moisture Anomaly (SMA) index (for an overview of existing drought indices refer to WMO and GWP, 2016). These standardized indices can be used for large-scale monitoring as they allow comparison in space. Often studies focus on one particular drought type or on the link between different types as droughts propagate, for example from precipitation deficit to soil moisture deficits and/or to surface and groundwater deficits (Haslinger et al., 2014; Bachmair et al., 2018, 2015; Blauhut et al., 2015; Stagge et al., 2015). A single variable assessment provides sector relevant information. Analyzing the root zone soil moisture drought signal for example provides information relevant for agriculture. However, different types of drought are not necessarily linked in a similar way in different catchments. For example, the propagation from meteorological anomalies to streamflow anomalies is affected by climate and catchment characteristics as well as by anthropogenic influences (Tijdeman et al., 2018). Catchments with high natural or artificial water storage might be able to sustain flow through short-term dry conditions, whereas catchments without significant water storage are likely more susceptible to short-term water deficits (e.g., Barker et al., 2016). Therefore, recent studies examine the simultaneous occurrence of different drought types (e.g., Brunner et al., 2019). In addition, below-normal anomalies in any hydro-meteorological variable do not necessarily lead to drought impacts on society and economy as impact occurrence and severity also depends on the vulnerability of a given system (Erfurt et al., 2019; Blauhut et al.; 2016, van Loon et al., 2016). While the hazard aspect of drought events is generally well understood, our understanding of vulnerability to drought is still limited. Also, comprehensive information on past drought impacts is still missing (van Lanen et al., 2016; Stahl et al., 2016; Kreibich et al., 2019). To fully understand past drought events, a multi-perspectives approach is essential. In particular, the development of plans to manage future droughts will benefit from synthesis and understanding of the complex patterns of past droughts across different sectors potentially impacted.

Knowledge on past extreme droughts is not only critical for future drought management but also for climate change research. As droughts are rare and irregular extremes, a valid analysis of extremes requires long-term records. Knowledge on past drought extremes can serve as a validation source for climate models and will increase the confidence in their projections of future changes in extremes. Due to the high natural variability in precipitation patterns, the influence of climate change on

future drought is still difficult to detect (Seneviratne et al., 2012). Knowledge of historical droughts can also help to assess the severity of current drought events (Hanel et al., 2018). For such comparisons, catalogues of drought events have been developed in many countries. For example, European summer droughts during the last two millennia were identified and catalogued based on tree-ring reconstructions by Cook et al. (2015) and Büntgen et al. (2010). A global database of meteorological drought events from 1951 to 2016 has been provided by Spinoni et al. (2019). For Ireland, a meteorological drought catalogue exists for the last 250 years derived using reconstructed precipitation series as well as documentary sources from newspaper archives (Noone et al., 2017) and for the Czech Republic drought events were catalogued using documentary evidence and meteorological records (Brazdil et al., 2013; Mikšovský et al., 2019). A systematic characterization of historical hydrological droughts (1891 to 2015) for a diverse set of catchments exists for the UK (Barker et al., 2019). Jakubínský et al. (2019) developed a repository of drought impacts (between 1981 and 2016) in the countries part of the Danube catchment. All these existing catalogues depict mainly one or two types of drought based on individual (e.g., precipitation, streamflow or tree-ring based) indices.

The purpose of this article is to catalogue and analyze historical drought events of the last 218 years at a regional scale, i.e. for the state of Baden-Wuerttemberg in southwestern Germany. Furthermore, this study explores multiple perspectives on drought and determines the added value of such datasets for a comprehensive understanding of drought events from the year 1801 onward. For this purpose, we:

a) combine information from four different sources: meteorological and hydrological observations, dendrochronological analysis of tree growth, and written evidence on past climate and drought impacts into a full catalogue of droughts,

b) compare the occurrence and assessment of major extreme drought events from these different perspectives,

c) investigate whether the recent clustering of extreme drought events is unprecedented in a historical context,

d) identify strengths and weaknesses of the datasets and indices, and

e) build a drought catalogue of "consensus-events" for the state of Baden-Wuerttemberg from the multi-disciplinary perspective.

## 2 Data and Methods

### 2.1 Multi-variable Dataset

The drought catalogue is based on datasets that represent the two major drought types - meteorological and hydrological drought as well as past drought impacts on vegetation and society. Based on these datasets, multiple levels of classification of individual and combined drought indices such as the SPI, the Standardized Precipitation Evapotranspiration Index (SPEI, Vicente-Serrano et al., 2010), and streamflow percentiles (QP) were calculated. Figure 1 gives an overview of the data, the derived indices and time series of drought events used (details in Section 2.2) and the overall analysis for the drought catalogue (Section 2.3). In this study, we use a compilation of several drought-related variables, herein referred to as datasets 1 to 4 (Fig. 1 and Fig. 2).

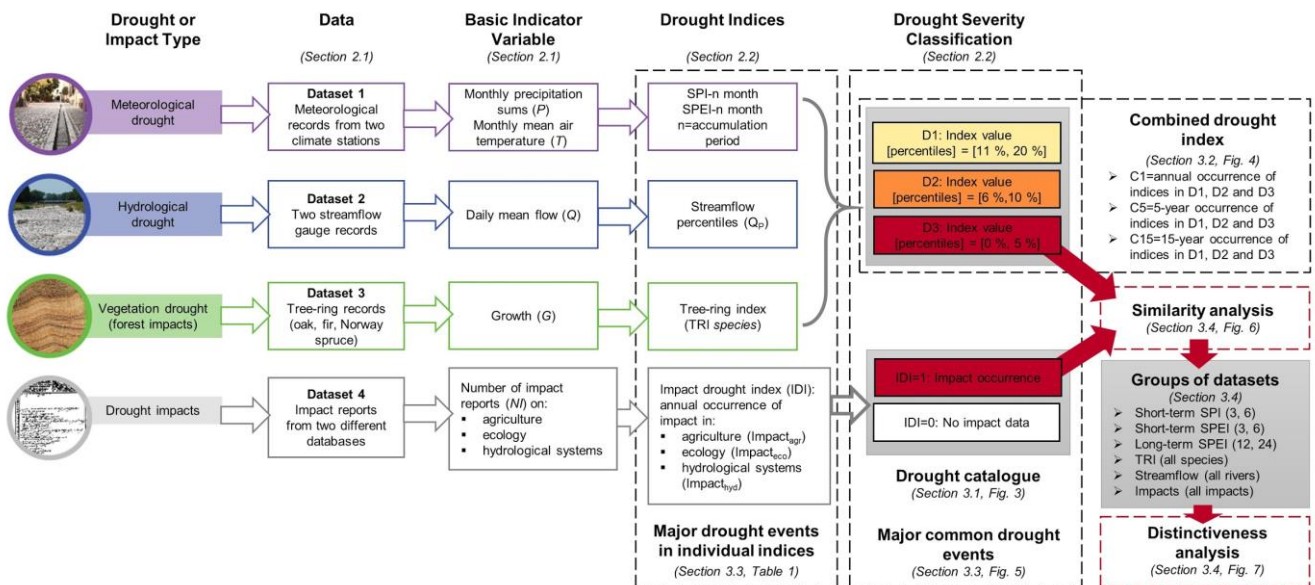

**Figure 1: Conceptual overview of the multi-perspective approach of this study. The used multi-variable dataset is shown as well as the derived indices used to create the drought catalogue.**

### 2.1.1 Dataset 1: Meteorological records

Dataset 1 comprises meteorological records from 1801 to 2018 of monthly mean air temperature ($T$) and monthly precipitation sums ($P$) for two stations in Baden-Wuerttemberg (Rheinstetten-Karlsruhe and Stuttgart, only the first is presented in this paper), which provided the longest continuous time-series of the required variables (Fig. 2). These $P$ and $T$ data were retrieved from the project "Historical Instrumental Climatological Surface Time Series Of The Greater Alpine Region" (HISTALP, Auer et al., 2007). All HISTALP series undergo homogenization procedures, which try to detect and eliminate non-climatic breaks and outliers (for more information see Auer et al., 2007). The time series of the HISTALP dataset cover the period 1801-2015 and were updated until the year 2018 using data from the German Weather Service (DWD, ftp://opendata.dwd.de/).

### 2.1.2 Dataset 2: Streamflow records

Dataset 2 consists of hydrometric records, namely daily streamflow records ($Q$) of the river Rhine at Basel and the river Danube at Kelheim (Fig. 2). Q data for the period of 1900-2018 were retrieved from the Bundesanstalt für Gewässerkunde (BfG, www.bafg.de). The selection of these two stations was based on the availability of long continuous datasets. The streamflow at Basel reflects mainly the alpine flow component into the study region and only to a lesser degree, the runoff produced within the study region. Nevertheless, the Rhine is an important river for the region of Baden-Wuerttemberg and its streamflow hence reflects whether social and economic impacts of drought are likely. The Danube River originates in the study region itself, i.e. its headwaters are in the Black Forest Mountains of southwest Germany, from where it flows eastwards cutting through limestone escarpment areas, and it also receives water from Germany's pre-alpine and alpine Region by way of its southern

tributaries. Both rivers are among Europe's longest. They support critical ecosystems and have played an important role for
the transport of goods, energy production, water supply and tourism for a long time.

### 2.1.3 Dataset 3: Tree-ring records

Dataset 3 consists of annually resolved tree-ring data from oak (*Quercus robur* L. and *Quercus petraea* (Matt.) Liebl.), fir
(*Abies alba* Mill.) and Norway spruce (*Picea abies* (L.) H. Karst.) trees from different sites in Baden-Wuerttemberg. These
tree-ring series span the past ~200 years. The combined tree-ring data stem from multiple sources. One dataset contains oak
tree-ring chronologies from the Rhine-valley which are described in more detail by Skiadaresis et al. (2019). Further, oak and
fir tree-ring series were provided by Büntgen et al. (2011) and Büntgen et al. (2014). Spruce and fir tree-ring chronologies for
the region were obtained from the international tree-ring database ITRDB (www.ncdc.noaa.gov) (Table S1). Finally, more
recent fir and spruce chronologies from the western part of the Black Forest were obtained from Schwarz and Bauhus (2019),
and from Sohn et al. (2013). We included only tree-ring series with a length of more than 40 years. To ensure a common signal
in each chronology, we restricted any further analyses to trees with high and significant inter-series correlation (IC) (IC > 0.3,
$p < 0.05$) with the respective mean chronology of each species or the regional mean chronology. The final dataset consisted of
2089 individual tree-ring series (1632 oak, 241 fir and 216 Norway spruce).

### 2.1.4 Dataset 4: Negative impact reports

Dataset 4 is based on reported textual information on the negative impacts of drought events from a variety of information
sources: Two existing databases, the collaborative research environment tambora.org (Glaser et al., 2015, 2013) and the
European Drought Impact Report Inventory EDII (www.geo.uio.no/edc/droughtdb/edr/impactdatabase.php, Stahl et al., 2016)
provided the starting point, but were amended to create the specific dataset used in this study. Historical information from
tambora.org comprises written documents from manifold sources and chronicles, flood marks and hunger stones, as well as
pictures and official records, newspapers and early numerical statistical records on harvest yields, food prices, ecological
impacts and societal information. The European Drought Impact Report Inventory (EDII) archives coded summaries of more
recent reports on negative environmental, economic or social drought effects. Historical information prior to 1900 stems from
tambora.org, more recent drought impact reports from the EDII. Impact reports from questionnaires and interviews in the EDII
were excluded because they merely focus on drought impacts on public water supply and hydropower production from the
year 2000 onwards. Some additional sources not embodied in either of the two databases were included as well (Nees and
140 Kehrer, 2002; Pfaff, 1846). All available information on reported impacts for southwestern Germany - from these databases
and the additional reports - were spatially referenced and time stamped. For the purpose of this study, all drought impact reports
were assigned to three impact categories (1) agriculture, (2) ecology incl. forests and (3) hydrological systems incl. e.g. water
use for drinking water and water-borne transportation (Table S2 lists the reclassification scheme of the EDII data).

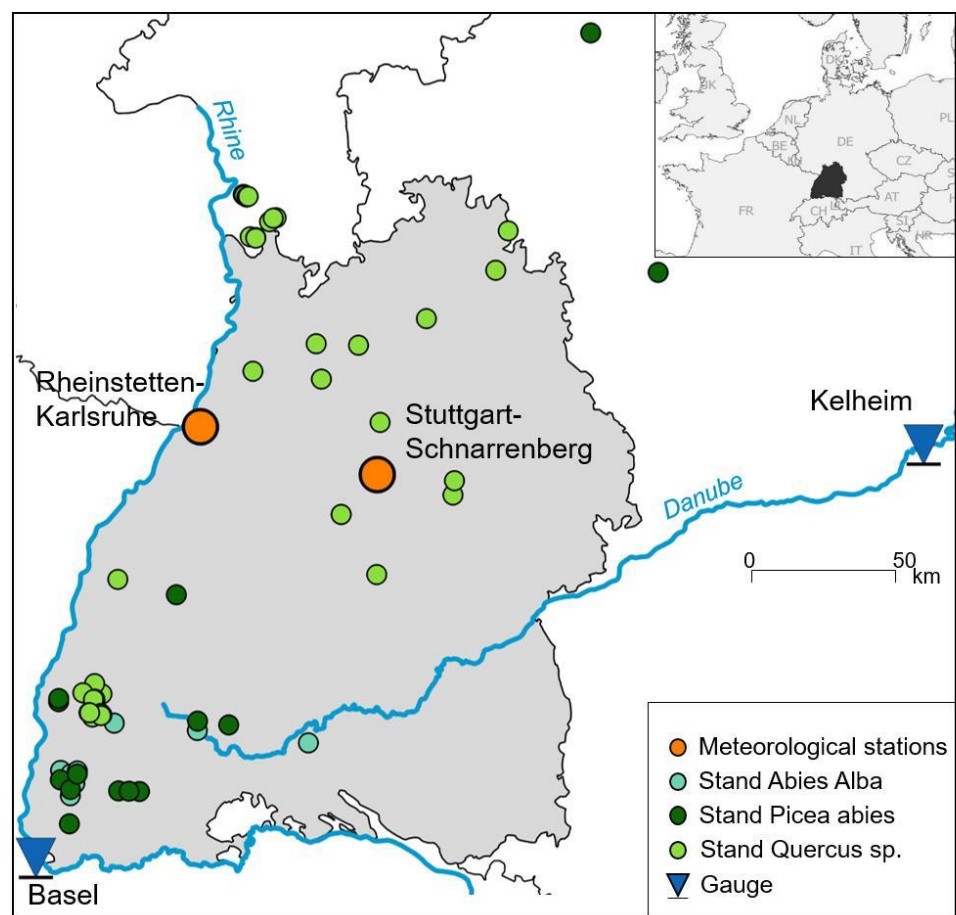

**Figure 2: Baden-Wuerttemberg and location of various data sources considered in the study.**

## 2.2 Drought indices and drought event severity classification

The four datasets were transformed into continuous time series of anomalies (Datasets 1-3) or impact occurrences (Dataset 4) in order to obtain drought indices that could be directly compared to each other (Fig. 1). The choice of variables and their transformations broadly followed common drought monitoring approaches such as the US Drought Monitor (https://droughtmonitor.unl.edu/). The indices were then used to explore the different characteristics and severity of past drought events for the different types of drought.

**2.2.1 Meteorological drought**

Meteorological drought, i.e. a lack of precipitation, was captured by the commonly used SPI and was calculated for Dataset 1. Given that summer droughts are often characterized by increased temperatures leading to an increased evaporative demand (e.g., Teuling, 2018), we also used the SPEI, which is based on the difference between precipitation and potential evapotranspiration. Potential evapotranspiration was estimated using the Thornthwaite equation which only requires

temperature and latitude as input, given that data to make more accurate potential evapotranspiration estimations (e.g., solar

radiation or wind speed) were not available for the entire period of record. In this study, we were interested in both short-term droughts during the summer and vegetation season as well as long-term droughts. Hence, the following accumulation periods (n) were chosen: SPI/SPEI-3 ending in June (early vegetation period important for tree growth) and August (summer period important for impacts), SPI/SPEI-6 ending in June and September (vegetation period important for tree growth), SPI/SPEI-12 ending in December (annual) and SPI/SPEI-24 (biannual) for December. SPI and SPEI were computed using the R Package "SPEI" (Version 1.7) from Beguería and Vicente-Serrano (2017). For the SPI calculation, a gamma distribution was fitted to the precipitation series between 1810-2018 to define the relationship between precipitation amounts and probability for given accumulation periods. Parameter estimation of the probability distributions was based on unbiased probability weighted moments (Beguería et al., 2014). The SPEI was calculated in the same way, but based on the climatic water balance (difference between potential precipitation and evapotranspiration). For the SPEI, standardization was based on the generalized logistic distribution (Beguería et al., 2014).

### 2.2.2 Hydrological drought

Hydrological drought was calculated from daily streamflow observations (Dataset 2). The daily streamflow *(Q)* data for the period between 1901-2018 was aggregated to annual as well as seasonal averages for both the non-winter (March-November) and the summer and autumn (June-November) seasons. The aggregated streamflow data were then transferred to streamflow percentiles ($Q_P$), using Weibull plotting positions $Q_P = \text{rank}(Q) / (n+1)$; where n in this case equals the amount of years (Weibull, 1938).

### 2.2.3 Vegetation drought

Vegetation drought indexing followed standard dendrochronological methods (e.g. Speer, 2010) in order to derive a tree-ring index (TRI) from Dataset 3. The 2089 different tree-ring series originated from 55 locations in Baden-Wuerttemberg (Fig. 2 and Table S1). To remove age-related growth trends from individual tree-ring series while maintaining their inter-annual variability, we detrended raw ring-width series using a 30 year spline with 50 % frequency response cutoff (the frequency at which 50 % of the amplitude of a signal is retained, see also Cook et al., 2013). This commonly used detrending approach removes the biological trends present in growth series (low frequency) while simultaneously preserving annual to decadal variability in growth (high frequency) (Cook and Peters, 1981; Speer, 2010). A bi-weight robust mean, which reduces the influence of outliers in the computation of the mean, was then calculated to generate four residual chronologies: oak, fir, spruce and a combined chronology including trees from all three species (see Fig. S1). The quality of the developed chronologies was assessed using several descriptive statistics (EPS: expressed population signal, SNR: signal to noise ratio, and rbar: mean interseries correlation) commonly used in dendrochronology (Speer, 2010; Table S1). The EPS is an indicator of how well a chronology represents a theoretical infinite population (Wigley et al., 1984). Low values of EPS (commonly <0.85) indicate that the chronologies are dominated by individual tree signals rather than a consistent regional signal (Speer, 2010). Rbar is the mean correlation between series within a chronology and is a measure of common signal strength of detrended chronologies. The SNR is a measure of the desired signal in each chronology versus the amount of unwanted information and

random variation (Speer, 2010; Cook et al., 2013). Quality control, detrending and chronology development were performed using the 'dplR' package in R (Bunn, 2008).

### 2.2.4 Socio-economic drought

Socio-economic drought was represented based on the reported drought impact information of Dataset 4. From this information on time, location, and impact category, an Impact Drought Index (IDI) was derived that contains an annual value indicating 'impact occurrence' (IDI=1) or 'no impact data' (IDI=0) in each category. As the number of impact reports changed over time with more digitized source material being available in the 20th and 21st century, we corrected for this trend in the data by converting every year before 1947 with one or more indicated impacts into an impact year with IDI=1. In the period 1947-1999, years with more than two reported impacts were characterized as impact years. After 2000, years with more than three reported impacts were considered as impact years.

### 2.2.5 Drought severity classification scheme

A drought severity classification scheme was then applied to the different individual drought indices to facilitate characterization and comparison of drought years across the continuous indices of datasets 1-3 (Fig. 1). A total of 12 meteorological (SPIn and SPEIn), six hydrological (QP) and four vegetation (TRIspecies) drought indices were considered for this analysis. Drought years were derived from the anomaly time series of the different indices (SPIn, SPEIn, QP, TRIspecies). A year was defined to be under drought whenever the variable of interest was abnormally low, in this study below the 20th percentile. Drought years were further classified according to three different severity classifications: D1 (moderate; <20th percentile), D2 (severe; <10th percentile) and D3 (extreme <5th percentile). Dataset 4 was classified according to years with impact occurrence (IDI=1) and years with no impact reports available (IDI=0).

Events identified using their severities comprise our drought catalogue: all drought events identified according to the individual indices (for data availability see Erfurt et al., 2020). Based on these severity classes the relative amount of all indices indicating drought was calculated, hereafter referred to as the combined drought index (C). Since the amount of available indices changed over time, C was calculated as the number of indices indicating droughts in certain years relative to the total number of available indices for that year. Different smoothing windows (1, 5 and 15 years, resp. C1, C5 and C15) were used to identify temporal clusters (Fig. 1). By using a moving window instead of fixed decadal time blocks, we assured that we did not miss decadal drought hot-spots that happen at the end of one decade and the beginning of the following decade.

To identify the major common drought events, the 20 drought years in which the most indices from different groups point to a drought were selected (Fig. 1). To compare and select the major drought years individually, we ranked the ten most extreme years since 1900 for all indices, and additionally since 1801 for the meteorological and the tree-ring dataset.

## 2.3 Analysis of similarities and differences in drought variables

Three metrics were used to assess similarities and differences in the ability of the multiple indices to identify drought events. The first metric assesses the full range of anomalies (dry and wet). These were quantified by Pearson's correlation coefficient ($r$) between all pairs of drought indices. As a second metric, we developed a similarity index ($s$) for extreme events only (D3).

A third metric was used to assess the differences in the identification of extreme droughts, which is hereafter called distinctiveness analysis.

The similarity index was calculated for each pair of drought indices. It relates the average number of extreme droughts in an index-pair to the number of common (simultaneous) droughts in the pair (Equation 1):

$$s = \frac{average\ number\ of\ extreme\ droughts}{number\ of\ common\ extreme\ droughts}$$

Similarity measures, *r* and *s*, were calculated for all drought indices for the whole period of record (1901 to 2011). In a second step, they were calculated for an earlier 40-year period (1901 to 1940) and for a later 40-year period (1972 to 2011) to assess

possible changes in the relationships between drought patterns in the different datasets over approximately the past 110 years. These periods were chosen for several reasons: first, we wanted to identify changes in extreme drought occurrence over time, and secondly, we wanted to include all groups of indices for this analysis. A reason to assess *s* for extreme drought events only was the hypothesis that global change may have intensified in particular the extremes in the more recent period.

To assess the distinctiveness of extreme droughts (D3) identified by the different datasets contained in the drought catalogue,

we first grouped the indices into different categories: (a) **Short-term SPI** (SPI-3 of June and August, and SPI-6 of June and September), (b) **Short-term SPEI** (SPEI-3 of June and August, and SPEI-6 of June and September), (c) **Long-term SPEI** (SPEI-12 and 24 of December), (d) **Tree-rings** (all four tree-ring chronologies), (e) **Low-flow** (Q-Rhine$_{(Mar-Nov)}$, Q-Danube$_{(Mar-Nov)}$, Q-Rhine$_{(Jun-Nov)}$ and Q-Danube$_{(Jun-Nov)}$) and (f) **Impacts** (agriculture, ecology and hydrology). Since there was almost no difference between the long-term SPEIs and the long-term SPI-12 and SPI-24 of December regarding the extreme droughts

we excluded the latter from this analysis. Then, we investigated which extreme drought events would not be identified if excluding a single index category. The number of extreme droughts that our final catalogue would have failed to identify had we not included a specific index category (e.g., tree-rings), was defined as its distinctiveness value.

## 3 Results

### 3.1 Drought catalogue: drought events according to individual indices

Figure 3 presents the temporal distribution of drought occurrence for the different indices and drought severities. Drought events of all severities (D1 to D3) occurred throughout the last 218 years (Fig. 3a). However, several years stick out in all datasets; these include e.g. 1842, 1865, 1993, 1921, 1947, 1949, 1964, 1976, 1991, 2003, and 2018. No temporal trends of drought occurrences in general could be detected, but several clusters of increased drought occurrence were identified over the last 218 years (e.g. 1860s and recent decade, Fig. 3a and c). Further, the occurrence of a drought event in a certain year often

is indicated by multiple indices. Especially the occurrence of extreme drought events is visible in most indices, whereas moderate droughts often only appeared in one (or few) of the considered indices.

The meteorological indices used in the catalogue identify the degree of dryness at different time scales (Fig. 3a). Droughts identified based on meteorological indices are fairly distributed over the study period, although especially the recent decade and the period between 1857 to 1870 show a higher severity of drought events. When taking potential evapotranspiration into account (SPEI for all different accumulation periods) in the recent decade more droughts are classified as extreme (e.g., 2011, 2015, 2018). For the year 1991, all meteorological indices denoted severe to extreme precipitation shortfall yet this translates only into a moderate drought intensity based on indices of tree-rings and streamflow.

The streamflow indices revealed that between 1976 and 2003, no severe or extreme drought was observed. For both the Rhine and Danube, mainly the streamflow in the years 2003 and 2018 were marked as extremely low. For the same years, all other variables apart from SPI-6 of June indicated a moderate to extreme drought year.

In many cases, dendrochronological records show that years of extremely low tree-growth coincided with drought events identified by other indices (Fig. 3a). For example, the years 1893 and 1976 are listed in the catalogue as extreme drought events based on all tree-ring chronologies and as extreme, severe or moderate droughts based on meteorological indices. On the other hand, in some cases tree-rings showed a delayed response to drought. For example, in 1921, radial growth of fir and spruce appeared to be not affected by drought as indicated by meteorological indices, but tree-ring indices of both species indicated a drought in the year afterwards (1922). All other indices marked 1921 as an extreme drought year. Meteorological indices revealed that the lack of precipitation occurred mostly before the summer season. SPI/SPEI-6 of September and SPI/SPEI-3 of August only show a moderate drought, whereas the lack of precipitation for the period March to June (SPI/SPEI-3 of June) of 1921 were classified as severe and for the half year January to June (SPI/SPEI-6 of June) as extreme droughts.

The drought impact time series provided insights into the actual occurrence of socioeconomic consequences of drought (Fig. 3b). In general, the drought events identified by impacts are in line with drought events, which were identified by meteorological indices. Remarkably, the years of 1853 and 1854 were not identified as drought events by the meteorological drought indices but were identified by impact reports. In 1853, reports focused on hydrological impacts such as extreme low flow in the Rhine and problems with public water supply (low groundwater levels). In 1854, Rhine streamflow was extremely low again, impacts on hydropower production were reported, and water levels of Lake Constance were reported to be the lowest on record.

The annual time series of joint drought occurrence according to different indices is shown in Fig 3c. This annual combined drought index (C1) was analyzed in order to explore the drought events from all perspectives (except the impacts) and examine how many D1, D2 and D3 droughts occurred in a specific unit of time (Fig. 3c). A cluster of extreme droughts between 2003 and 2018 was identified by all indices, with the SPEI accounting for the largest share. Through the analysis of a long-term dataset reaching back to 1801, it becomes evident that the cluster of recent droughts are not exceptional. From the late 1850s to the early 1870s clusters of extreme droughts occurred (Fig. 3c). Another cluster was found at the end of the 1940s. In addition, the 1960s are marked by several consecutive drought years. Years with more than 25% of the indices pointing to an extreme drought event are 1842, 1865, 1893, 1921, 1949, 1964, 2003, 2015 and 2018.

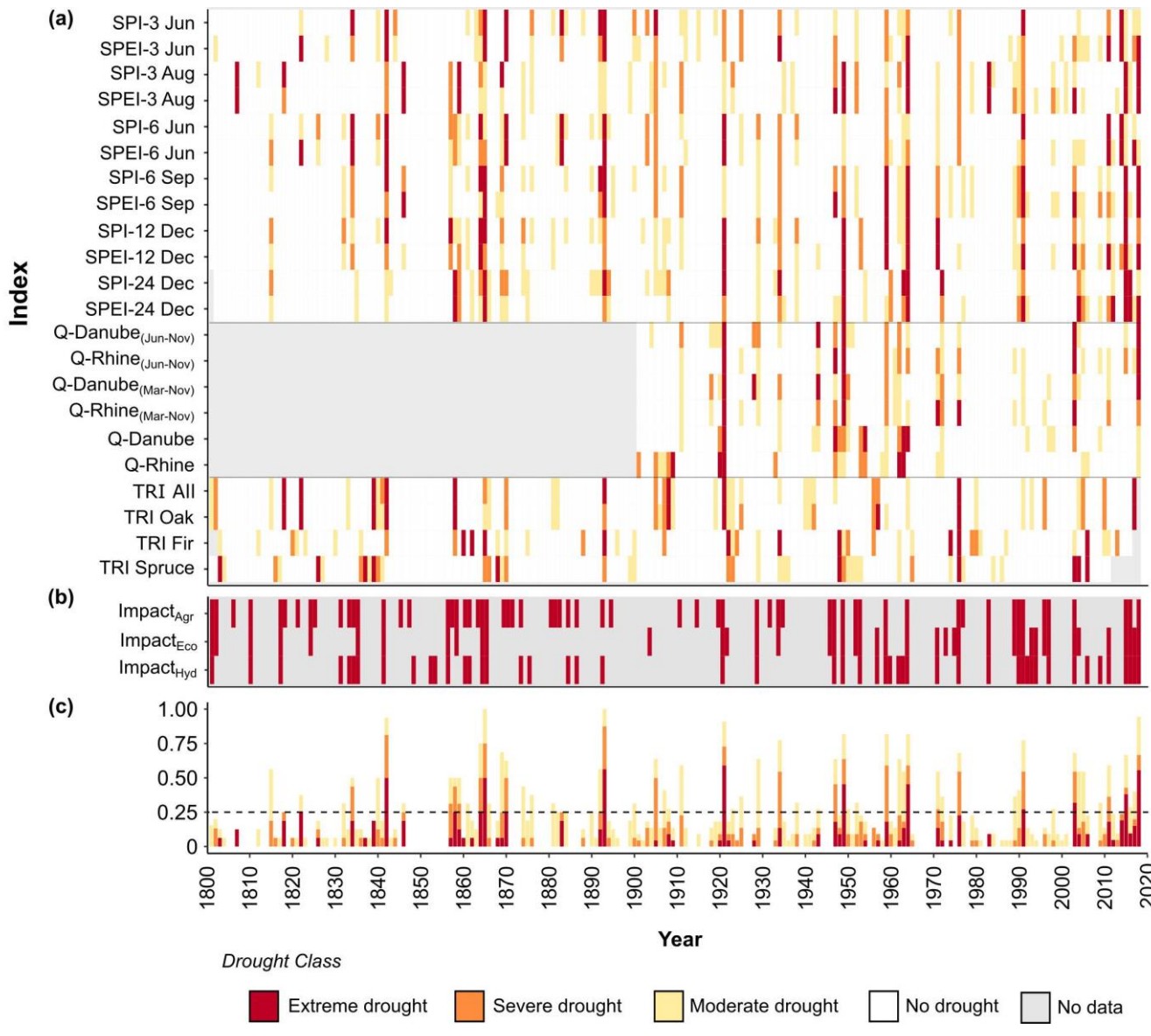

**Figure 3: Annual time series of drought occurrence in southwestern Germany according to different groups of indices (a) hydro-meteorological drought indices and tree-ring data, (b) drought impacts, and (c) the combined drought index C1, which shows the annual percentage of indices indicating droughts (impacts are excluded).**

## 3.2 Combined drought occurrence

To survey if drought occurrence has changed over time since 1801, the combined drought index for 5 (C5) and 15 (C15) years-long time windows was used to identify temporal patterns of drought occurrences (Fig. 4). The combined occurrence describes the average percentage of indices classified as drought in a 5 or 15 year window prior to the year of interest. Both analyses reveal a high overall variability over the last 218 years. If using a 5 years-long window "drought hotspots" which occurred in the 1860s and in the most recent decade are revealed (Fig. 4a). The analysis with a 15 years-long window (Fig. 4a) picked up more long-term variations in drought occurrence.

According to the five year rolling window, most droughts classified as extreme were found between 2014 and 2018 (last bar in Fig. 4c). The fraction of extreme drought occurrence in these five years stands out when compared to any other 5-year window since 1801. Nevertheless, the clustering of several years with a high occurrence of moderate, severe and extreme droughts is not unprecedented in the study area. During the decade from 1847 to 1857, no droughts occurred (Fig. 3a and Fig. 4b) while shortly after, 1860 and 1870, a high number of extreme droughts in tree-rings and meteorological datasets were identified (Fig. 4b). A major peak in drought occurrence was reached in 1865. A comparable increase of droughts was observed between the 1950s and the 1970s, with a peak around 1964. Also in the 1920s and towards the end of the 1940s the 5-year occurrences were high.

The combined drought index of a rolling 15-year window shows long-term periods of drought occurrence and absence (Fig. 4a). An increase of droughts was detected towards the 1840s. Most drought-prone periods were around 1870, the 1960s and with a steady increase in extreme droughts in the last decade.

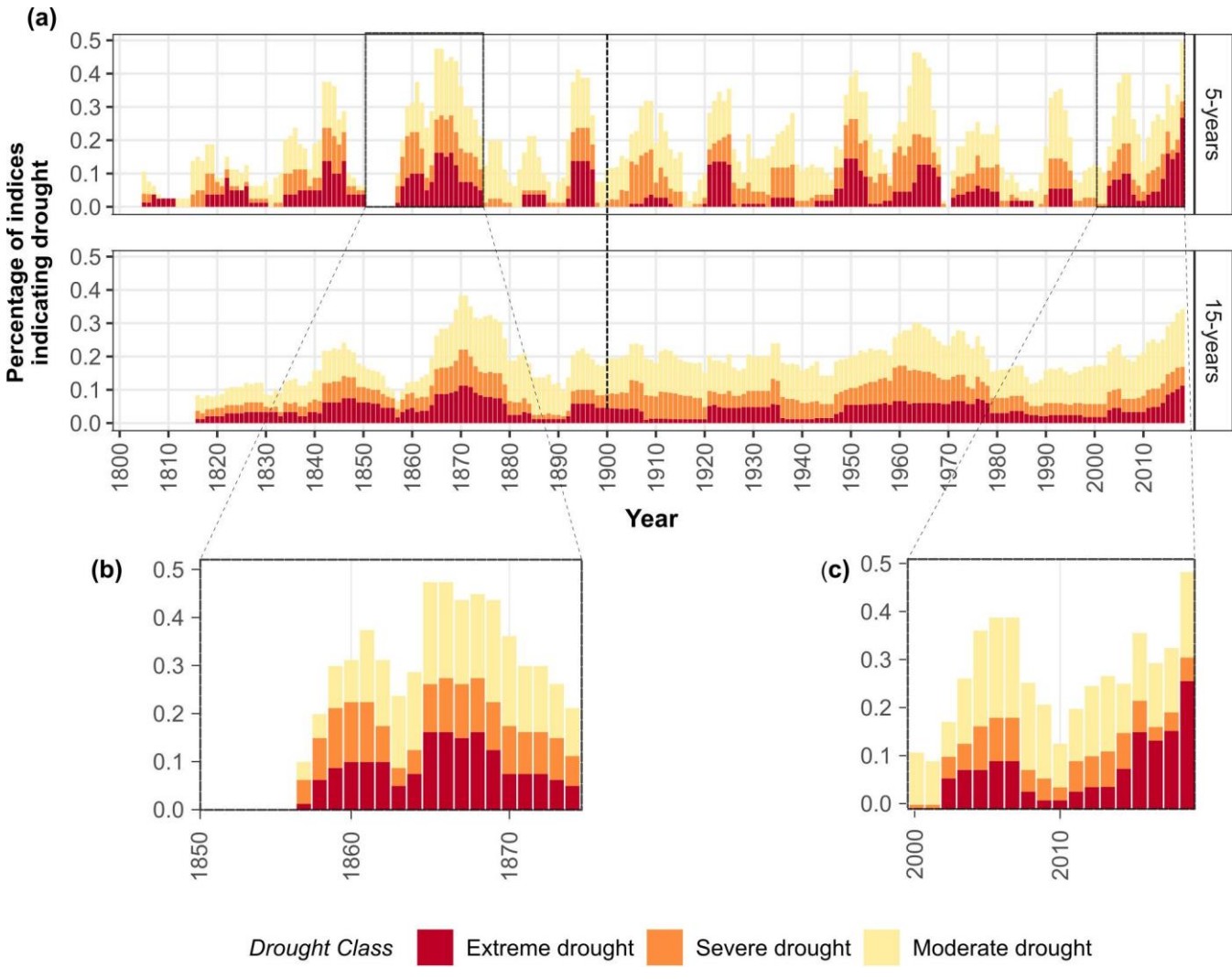

**Figure 4: The combined drought index C5 and C15, showing the relative amount of all indices in drought (different severities) smoothed with a 5 (C5) and 15 (C15) year backward smoothing window.**

### 3.3 Common drought events from different perspectives

To explore the different dimensions of droughts, we focused on the 20 years, for which more than ten indices point to a drought of any severity (Fig. 5). In all cases, more than 25 % of the indices were classified as a moderate, severe or even an extreme drought (Fig. 3c). In all these years, drought appeared in both the indices and the impacts. After 1900, all 17 years (the period for which streamflow data was available) were identified as drought years based on streamflow indices. In 1911, 1971 and 2015 a drought signal was not observed in the tree-ring indices but in most other indices. Although we used a large number of

meteorological drought indices (SPI and SPEI), these extreme years including 1921, 1976 and 2003, were identified in other index categories more or less equally (Fig. 5).

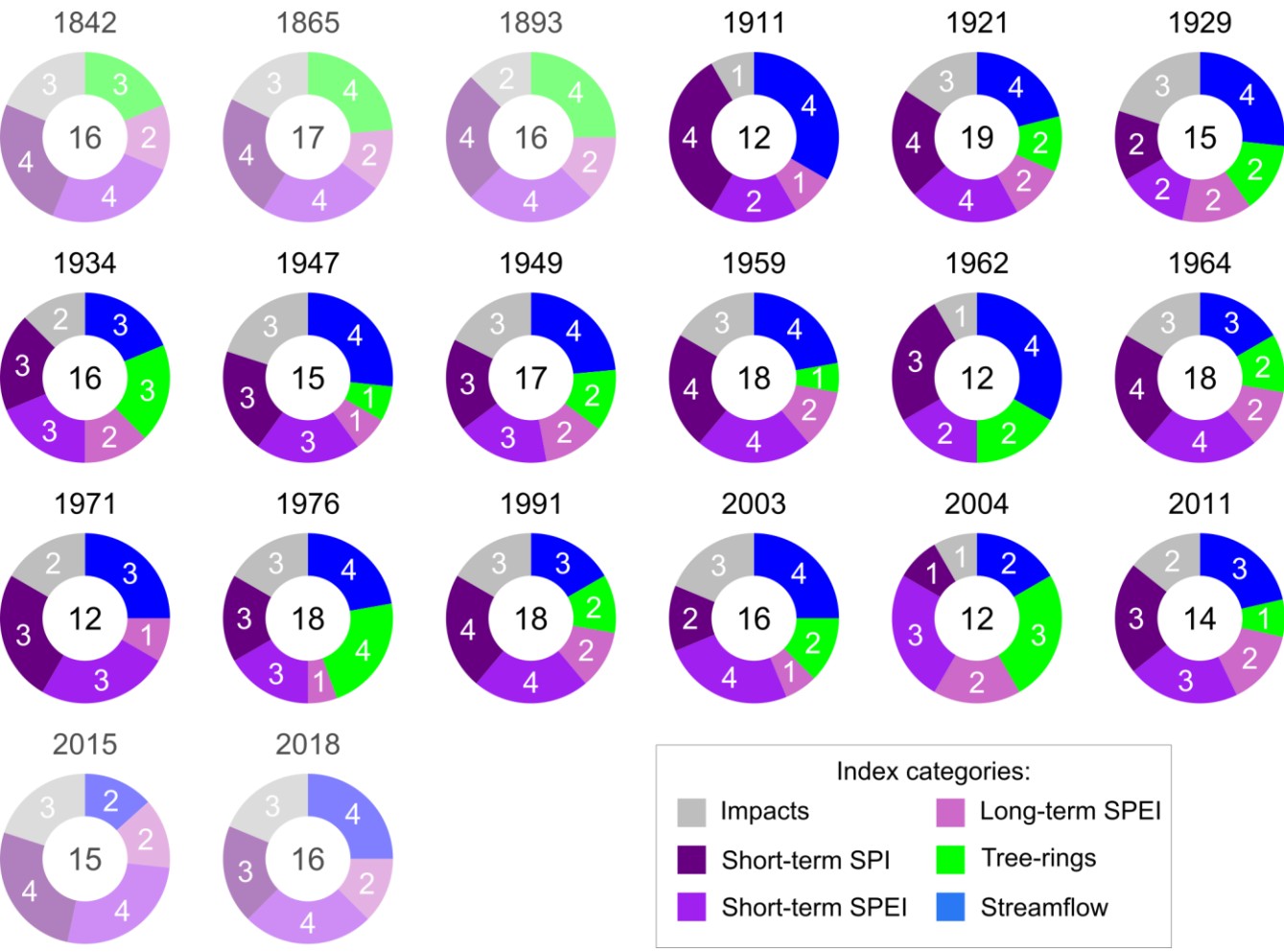

**Figure 5: Drought years in which more than ten indices (from different categories) indicate a drought (the total number of indices pointing to a drought is written in the circle, the number of indices per group pointing to a drought is written in each section of the ring). Years without streamflow data (prior to 1900) or without tree-ring data (after 2011) are displayed with paler colors.**

The ten most extreme drought years according to each index showed a large variation among indices (Table 1). Regarding its severity, the drought year 2018 ranked among the most extreme droughts in the last 118 years based on several indices (Table 1a). For 11 out of the 18 indices, 2018 was counted as a top ten event. Also 2003 stood out, in particular according to both meteorological and hydrological drought indices. In nine out of the 18 indices, 2003 was counted as a top ten event. Considering two consecutive drought years (SPEI-24 of December), six of the top ten droughts occurred within the last 18 years. For the other meteorological indices, two to four droughts that occurred after the year 2000 were classified among the top ten. Events in the 1940s (1947 and 1949) ranked in the top ten across the meteorological and hydrological drought indices.

In the tree-ring dataset, only spruce was affected in the same year. Other extreme drought events occurred in 1921, 1971 and
1976. For tree-rings, especially the years 1921 and 1976 stood out.

When ranking the last 218 years, which is possible for the meteorological and tree-ring dataset, many of the top ten droughts occurred in the 19th century, e.g., 1842, 1865 and 1883 (Table 1b). More than half of the top ten drought events, identified by short-term SPI and SPEI, occurred in the 19th century. Only for SPEI-6 of September and the long-term SPEIs (12 and 24 of December), less than half of the top ten events occurred in the 19th century. For these three indices, 2018 remains still the top

one event (Table 1b). Also the drought events in 2003 and 2015 were still among the top ten for the majority of meteorological indices.

**Table 1: Overview of the top ten most severe drought years. Drought years since the year 2000 are colored purple and years between 1801 and 1900 are marked green. * = years with the same value as the year before. "-" = no data available.**

**(a) Top ten since 1900.**

| Rank | SPI-3 Jun | SPEI-3 Jun | SPI-3 Aug | SPEI-3 Aug | SPI-6 Jun | SPEI-6 Jun | SPI-6 Sep | SPEI-6 Sep | SPEI-12 Dec | SPEI-24 Dec | TRI All | TRI Oak | TRI Fir | TRI Spruce | Q-Rhine(Mar-Nov) | Q-Danube(Mar-Nov) | Q-Rhine(Jun-Nov) | Q-Danube(Jun-Nov) |
|---|---|---|---|---|---|---|---|---|---|---|---|---|---|---|---|---|---|---|
| 1 | 1905 | 2011 | 1983 | 1983 | 2011 | 2011 | 2018 | 2018 | 2018 | 2015 | 1976 | 1921 | 1976 | 2003 | 1921 | 1921 | 1949 | 2018 |
| 2 | 1991 | 2014 | 1949 | 2018 | 1991 | 2014 | 1959 | 2003 | 2015 | 2016 | 1921 | 1976 | 1922 | 1948 | 1949 | 1949 | 2018 | 1949 |
| 3 | 2014 | 2018 | 2018 | 2015 | 2014 | 2017 | 2015* | 1947 | 1959 | 2018 | 2017 | 2017 | 1948 | 2004 | 1976 | 2018 | 1947 | 1921 |
| 4 | 1934 | 1964 | 1964 | 2003 | 1921 | 1921 | 1964 | 2015 | 2003 | 1964 | 1908 | 1908 | 1974 | 1976 | 2003 | 1943 | 2003 | 1943 |
| 5 | 1925 | 1934 | 2015 | 1947 | 1903 | 1959 | 1991 | 1964 | 1971 | 2004 | 2010 | 1957 | 2006 | 2006 | 1971 | 1928 | 1921 | 2003 |
| 6 | 2011 | 1905 | 1947 | 1964 | 1929* | 1991 | 1947 | 1959 | 1949 | 1991 | 1956 | 1905 | 1934 | 1922 | 2011 | 1934 | 1964 | 1911 |
| 7 | 1964 | 1976 | 1962 | 1949 | 1905 | 1934 | 1905 | 1991 | 1921 | 2012* | 2005 | 1956 | 1956 | 1949 | 2018 | 1950 | 1976 | 1962 |
| 8 | 1921 | 1947 | 1905 | 1991 | 1938 | 1976* | 1971* | 1949 | 1964 | 1949 | 1905 | 2010 | 2013 | 1923 | 1947 | 1972 | 1959 | 1928 |
| 9 | 1900 | 1925 | 1991 | 1976 | 1959 | 1905 | 1949 | 2011 | 1991 | 2005 | 1907 | 1942 | 1907 | 1965 | 1943 | 1918 | 2015 | 1947 |
| 10 | 1976 | 2017 | 1911 | 1905 | 1934 | 1964 | 1911 | 1971 | 2011* | 1921 | 1996 | 2005 | 1950 | 1962 | 1972 | 2003 | 1971 | 1959 |

**(b) Top ten since 1801.**

| Rank | SPI-3 Jun | SPEI-3 Jun | SPI-3 Aug | SPEI-3 Aug | SPI-6 Jun | SPEI-6 Jun | SPI-6 Sep | SPEI-6 Sep | SPEI-12 Dec | SPEI-24 Dec | TRI All | TRI Oak | TRI Fir | TRI Spruce | Q-Rhine(Mar-Nov) | Q-Danube(Mar-Nov) | Q-Rhine(Jun-Nov) | Q-Danube(Jun-Nov) |
|---|---|---|---|---|---|---|---|---|---|---|---|---|---|---|---|---|---|---|
| 1 | 1870 | 1870 | 1983 | 1983 | 1870 | 2011 | 2018 | 2018 | 2018 | 2015 | 1822 | 1822 | 1976 | 2003 | - | - | - | - |
| 2 | 1893 | 1865 | 1949 | 2018 | 1842 | 2014 | 1865 | 2003 | 2015 | 1865 | 1858 | 1858 | 1922 | 1948 | - | - | - | - |
| 3 | 1842 | 1893 | 1846 | 1846 | 1883 | 1870 | 1893 | 1947 | 1959 | 2016 | 1976 | 1921 | 1893 | 2004 | - | - | - | - |
| 4 | 1905 | 2011 | 2018 | 2015 | 1893 | 1893 | 1959 | 2015 | 2003 | 2018 | 1921 | 1976 | 1948 | 1837 | - | - | - | - |
| 5 | 1865 | 2014 | 1818 | 2003 | 2011 | 1842 | 2015* | 1865 | 1971 | 1964 | 2017 | 2017 | 1862 | 1976 | - | - | - | - |
| 6 | 1883 | 2018 | 1964* | 1807 | 1991 | 1834 | 1964 | 1964 | 1949 | 2004 | 1842 | 1842 | 1860 | 1803 | - | - | - | - |
| 7 | 1991* | 1822 | 1859 | 1947* | 1834 | 1822 | 1892 | 1959 | 1921 | 1991 | 1839 | 1908 | 1974 | 1839 | - | - | - | - |
| 8 | 1834 | 1842* | 1807 | 1964 | 1864* | 2017 | 1991* | 1991 | 1964 | 2012* | 1908 | 1839 | 2006 | 1826 | - | - | - | - |
| 9 | 2014 | 1964* | 1869 | 1859 | 2014 | 1921 | 1842 | 1846 | 1864 | 1949 | 1893 | 1818 | 1865 | 1868 | - | - | - | - |
| 10 | 1892 | 1934 | 2015 | 1949 | 1921 | 1883 | 1864 | 1893* | 1991 | 1858 | 1818 | 1957 | 1934 | 2006 | - | - | - | - |

## 3.4 Similarities and distinctiveness of drought in different datasets

The similarity index, calculated for each pair of indices, showed some interesting patterns in the extreme droughts identified (Fig. 6). Streamflow percentiles from the two rivers and for the two different accumulation periods (March to November and June to November) were very similar in terms of identified drought events in the early period (1901 to 1940). In the later period (1972 to 2011), the occurrence of below normal streamflow coincided less often in both rivers. The years classified as extreme drought events by the tree-ring chronologies of oaks and of combined tree species were identical for both periods. However, in the early period the two conifer chronologies (spruce and fir) did not show any similarity with the combined or the oak chronology in identifying drought events. On the contrary, in the later period approximately half of the identified droughts were commonly classified as extremes in all three different tree-ring chronologies. With a few exceptions (SPEI-6 of June – SPEI-12 December, SPI-6 of June – SPEI-12 December, SPEI-6 of June – SPI-6 of June), meteorological drought indices were very dissimilar in identifying drought events during the early period. Their similarity however increased in the later period. The SPI/SPEI-3 classified only a single extreme drought in the early time period. Therefore, for this time period no similarities existed with other datasets. Streamflow and SPI-6 of June, SPEI-6 of June, and SPEI-12 of December showed almost identical extremes in the early period. Their similarity became weaker in the late period in the case of SPEI-12 December while no similarities were observed between streamflow series and SPI-6 of June or SPEI-6 of June. Low-flow series showed some similarity with SPEI-3 of August in the late period (except for Q.Danube Mar-Nov) which was not observed in the earlier period.

Relationships among different indices as well as temporal changes in their relationships computed by Pearson's correlation coefficients among all pairs of datasets and for two 40-year periods are presented in detail in the supplementary material (Fig. S2 and Fig. S3). Several correlations between the different indices over the whole period (1901-2011) and for two shorter time periods (1900 to 1940 and 1972 to 2011) were observed (Fig. S3). As expected, the strongest correlation was found between indices belonging to the same type of dataset or time-scale (short-term SPI, short-term SPEI, long-term SPEI, tree-rings and streamflow) for both investigation periods. The two meteorological drought indices (SPI and SPEI), calculated for different accumulation periods and ending in different months, correlated strongly with each other (Fig. S3). In addition to the expected relationships between indices belonging to the same group, strong correlations were also observed between indices belonging to different groups: Streamflow percentiles correlated most strongly with long accumulation periods (12 and 24 months) of meteorological indices both in the early and later period ($r > 0.6$). Tree-ring chronologies showed overall weak correlations with streamflow anomalies with the exception of oak and the combined tree-ring chronologies, which were significantly correlated with the two streamflow series from the Rhine river. However, the combined tree-ring chronology as well as the oak chronology showed strong positive correlations with short-term meteorological drought indices in both periods (Fig. S2 and Fig. S3).

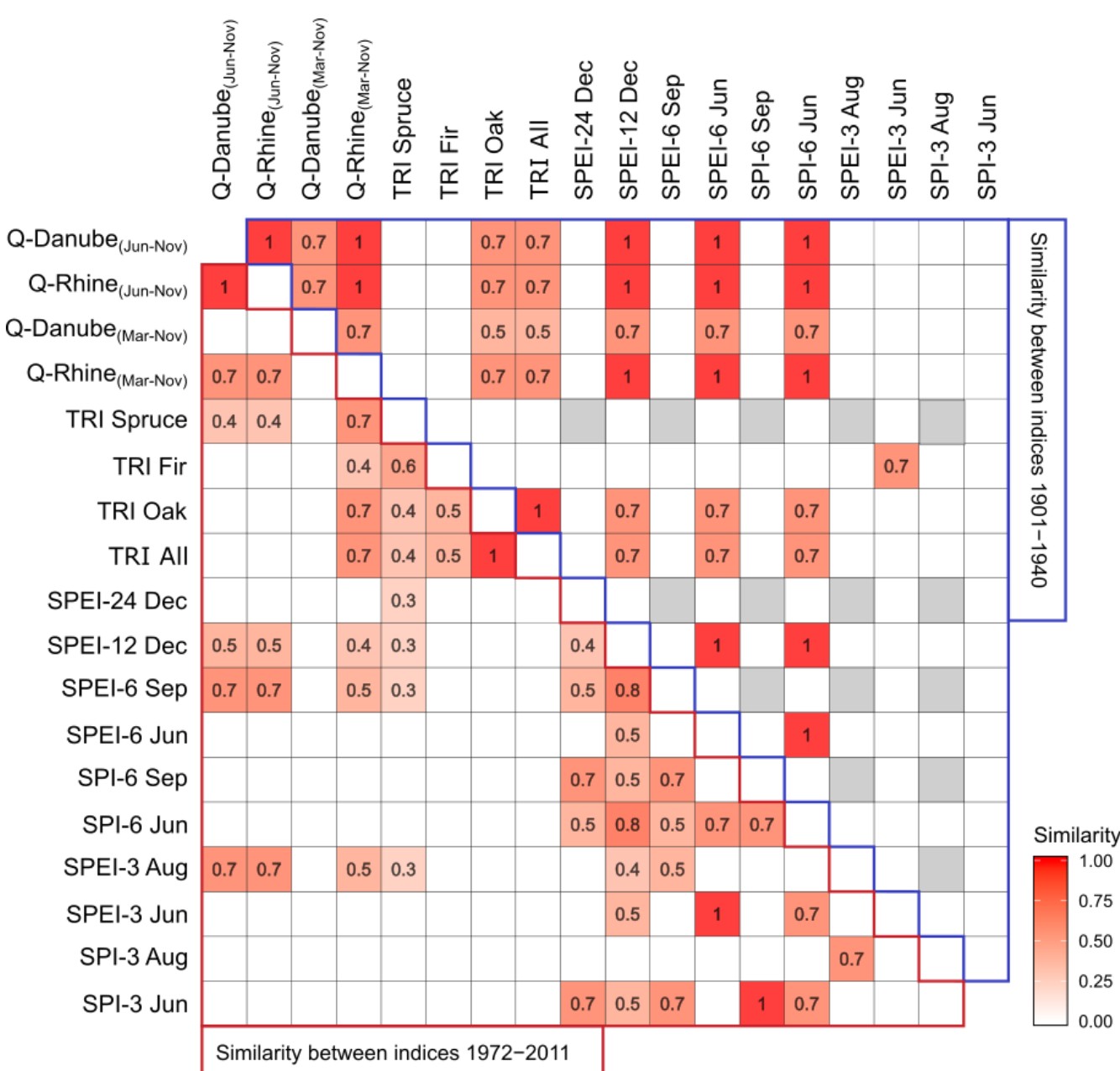

**Figure 6: Similarity index (*s*) between the different pairs of drought indices for two different time periods (1901 to 1940 and 1972 to 2011). Grey boxes = no extreme drought according to both indices.**

The distinctiveness value for groups of indices (Fig. 7) allowed to determine drought events we would have missed had we

not included a specific type of dataset in the analysis. Including drought indices and impacts derived from different datasets in

our drought catalogue resulted in the classification of 57 years throughout the period between 1800 and 2018, for which at

least one drought index indicated extreme drought conditions (Fig. 3 and Fig. 7). Almost half of these drought events were identified by a single group of datasets. Tree-rings showed the largest number of unique droughts (12 in total) while drought impact datasets identified seven extreme drought events that were not characterized as extreme in the other groups of datasets. Interestingly, 13 out of the 19 extreme droughts that we would have missed had we excluded both tree-ring and impact data appear in the 19th century, while only three distinct droughts were observed in tree-ring and impact data after 1950. All groups of meteorological droughts had very low distinctiveness values (numbers of droughts excluded in Fig. 7) with short-term SPEI (accumulation periods of 3 and 6 months) showing no unique extreme drought events. This analysis indicates that different datasets provided distinct information on drought, which would be missed if only using one of the datasets.

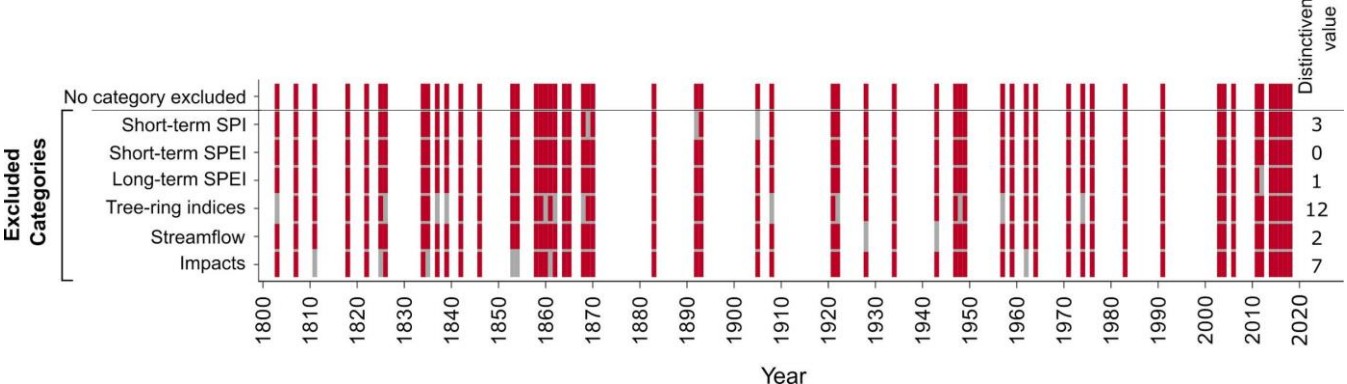

**Figure 7: Extreme drought events identified (red) or missed (grey) when excluding a certain category of drought indices. Each row shows the droughts that are identified when one of the categories are excluded from the analysis and therefore show the events unique to this category. The total number of missed events (distinctiveness value) when excluding a certain group of indices is displayed on the right.**

## 4 Discussion

### 4.1 Assessment of the drought catalogue and underlying data

With the drought catalogue a unique dataset was created, which comprehensively identified drought events since 1801 in southwestern Germany from multiple datasets and drought indices. Events occurred in all decades, but a particular clustering was found in the mid-19th Century, some consecutive years of extreme droughts in the early 20th Century and during the most recent decade. Many of the major droughts that were apparent across all groups of indices, were also identified by other studies focusing on different European regions and using different datasets. Extreme droughts from the 19th century were also identified by Brazdil et al. (2019) for the Czech Republic (e.g., drought of 1842) using documents and measured data. The extreme droughts in 1842, 1834, 1858, 1864, 1865 and 1893 were also identified in the Old World Drought Atlas by Cook et al. (2015), which provides summer (June-August) reconstructions of the self-calibrating Palmer Drought Severity Index (PDSI), based on tree-ring data. Increased drought occurrence in the years shortly before and after 1865 was also detected for an alpine-drought study by Haslinger et al. (2018) using meteorological data. The extreme drought events of the 21st century

were also detected by several studies on a European scale (e.g., Brunner et al., 2019; Lahaa et al., 2017; Ionita et al., 2017; van Loon et al., 2017). These findings relate to the fact that especially persistent extreme droughts occur due to global weather patterns, dictated by ocean and land temperature fluctuations (Ault 2020).

All data sources used have some uncertainties and hence, our results reflect some of the methodological choices made in this study: For example, we used hydro-meteorological averages aggregated over periods that we assumed are relevant for various drought related impacts, and compared those averages among the different years. However, using different aggregation periods or different methods (e.g., annual minimum flow or duration of which flow was below a fixed threshold) might have changed the ordering and classification of drought events. In addition, the long-term meteorological records of the stations Karlsruhe and Stuttgart are representative for the areas near these major civil centers for which drought would have had relevant impacts over the entire period of record. However, these meteorological records are not necessarily representative for the entire state of Baden-Wuerttemberg, which has a more diverse climate (especially in the more remote areas and at higher elevations). Furthermore, the rivers Rhine and Danube are major rivers flowing through the study area, which were important for, e.g., several economic activities. However, these larger rivers are not necessarily representative for the variety of streamflow regimes found for the smaller tributaries in the study area. In addition, streamflow, especially for larger river basins, is affected by direct and indirect human activities and thereby susceptible to changes over time (Tijdeman et al., 2018).

Tree-ring records were a valuable source of information regarding the impacts of drought on forests. Indeed, most years that were identified by meteorological or streamflow drought indices as severe or extreme droughts coincided with extremely reduced annual growth for all three species included in our analyses. Interestingly, we also observed unique differences in the drought response among some of the species. For example fir and spruce showed a delayed response to the drought in 1921, which was identified as drought by all other indices including the oak chronology. However, a detailed discussion of such differences in the timing of growth responses of trees to drought is beyond the scope of this study (but see for instance the study by Bhuyan et al., 2017 and discussion in Büntgen et al., 2010). In any case, delayed growth responses should be assigned to the drought years that triggered them so that they can be used for multi-index analyses such as presented here.

Finally, the drought impact information used in this study stems from two different databases that originally were created for different purposes. Although care was taken when simplifying the impact dataset into three categories, the results of the identified droughts in the two centuries should be regarded with this difference in mind. Historical reports from tambora.org stem from a rather limited variety of sources such as city chronicles or analog statistical yearbooks and mostly focus on food security, economic losses or impacts on human health (Erfurt et al., 2019). From the 20th century onward, novel media such as newspapers but also scientific publications became a source of information. The more recent years are dominated by a wealth of online mass media and a strong increase in drought research, which now also considers ecological impacts (Blauhut

et al., 2016, 2015b). The many impacts reported in recent drought years (e.g., 2003, 2015, 2018) likely also reflect an increased
awareness towards changing climate and changing policies, including climate change adaptation.

## 4.2 The added value of a long-term dataset

Investigating the development of drought occurrence and severity in the long-term (1801-2018) with independent observation data allowed an assessment of the uniqueness of the recent drought events. Investigating the top ten drought events since 1900 and since 1801 emphasized the importance of using datasets that date back as far as possible in order to contextualize today's event severity and frequency. When only considering the past 50 to 100 years, the years 2003, 2011, 2015 and 2018 were among the most extreme events, especially according to the drought indices that consider temperature (Table 1a). However, by considering the past 218 years, it becomes evident that drought events of similar severity (e.g., 1842, 1865, 1870, 1893) also did occur in earlier years (Table 1b).

Two main periods, one in the 19th century (1857-1870) and a second in the 21st century (2003-2018), of increased occurrence and severity were identified in our dataset (Fig. 4b and c). The result highlights that the recent frequent occurrence of droughts in the study area is not unprecedented when looking back into the 19th century. This highlights the need for long-term data, especially when studying the development of drought trends in the future. Although these two periods of frequent drought events were similar to each other in terms of overall percentage of indices pointing to drought, the recent period (starting in 2003) was characterized by a larger number of extreme droughts. This rise in severity in recent years is a result of increasing precipitation deficits and rising temperatures (Hänsel et al., 2019; Dai, 2013). At a European scale, studies have shown trends towards more droughts in Southern Europe and wetting trends in Northern Europe (e.g., Gudmundsson and Seneviratne, 2015; Vicente-Serrano et al., 2014), whereas for Central Europe the detection of drought trends is still an ongoing discussion (Seneviratne et al., 2012).

## 4.3 The added value of a multidisciplinary dataset

The inclusion of different drought indices and impact information in this study allowed a comprehensive assessment of past extreme drought events based on observations. Based on all indices, 57 out of the past 218 years (26 %) were characterized as extreme drought events in the region by at least one of the included indices (Fig. 3). This result highlights the uniqueness of different drought years. The analysis of different categories of drought indices further showed some extreme droughts that we would have missed when using only one category. For example, some years with extreme precipitation deficits and concurrent negative impacts for society were correctly detected by meteorological indices but not identified as extreme by tree-rings or streamflow data. Using a multidisciplinary dataset thus provided a more complete picture of droughts in the past.

In the present study, several drought events might have been overlooked if we had excluded any of the assessed indices (Fig. 7). Interestingly, we observed a change in the number of unique extreme droughts (number of extreme drought events missed

(grey) when excluding any category of indices in Fig. 7) over the last 200 years. The majority of these distinct droughts were
identified by either tree-ring or impact data and appeared before 1900, while only three distinct events were found based on
these indices after 1950. One possible explanation for the decrease in distinct events could be an improved quality and accuracy
of instrumental records in Germany, especially after the end of World War II.

## 4.4 Value of drought catalogue for drought management

The integration of long time series showed the non-uniqueness of the recent drought period in time. For drought monitoring,
going back further in time a longer reference has the advantage of placing ongoing events in a longer context of occurrence
frequency. As a result, the recent droughts may not appear as severe as in studies with shorter term reference periods. As
extremes are by definition, rare events, knowing more about historic events that could be used as "design events" (see e.g.,
Stoelzle et al., 2018) can help improve both short- and long-term drought planning and management. As a result of this long-
term catalogue, we found that multi-year droughts may be a type of event that should be better prepared for even in the study
area's temperate humid climate.

The unique drought events identified in the different indices of our drought catalogue reveals the added value of considering
different variables. Including drought impact information provides the necessary context to evaluate the severity of historic
drought events, especially given the large changes in vulnerability over the past century. As Erfurt et al. (2019) have pointed
out for the state of Baden-Wuerttemberg, the strength of a drought impact and the societal consequences strongly depend on
the societal vulnerability and resilience as well as the possibilities to cope with and adapt to these impacts. The drought period
identified in the mid-19th century (Fig. 4b), was also a time of growing population and political changes. As the historical
records underlying our dataset show, instability as well as drought relevant harvest failures and pricing led to hunger and
diseases as well as an increase in mortality. Similarly, the drought events in 1921-22 and 1947-49 were characterized by
increased vulnerability. The damages and losses after the First and Second World War had increased the sensitivity towards
drought related impacts. The recent drought of e.g., 2015 and 2018 in southwestern Germany that impacted multiple sectors
(e.g. forestry, agriculture, energy and industry) could be tracked via some of the existing drought monitoring products available
for the region. The State Institute for the Environment of Baden-Wuerttemberg (LUBW) offer a basic low flow monitoring
system (www.hvz.baden-wuerttemberg.de) and the DWD provides monthly maps of an aridity index for Germany
(https://www.dwd.de/EN/ourservices/klimakartendeutschland/klimakartendeutschland.html). However, the scarce amount of
information is scattered around different places and there is no centralized drought management authority. The presented
results suggest the added value of having multi-variable long-term drought information available in a central platform.

## 5 Conclusions

The main objectives of our analyzes were a) to learn more about droughts in the region from a long-term perspective and b) to
conduct a multidisciplinary analysis of drought events across various sectors using different datasets and hence combine

knowledge from different disciplines. The drought catalogue provides valuable information on long-term drought occurrence in southwestern Germany, which can be expanded in further drought studies. Our long-term dataset reveals that recent drought clusters are less exceptional in a historical context than when looking at the last 30-40 years as often done in trend analyses. Analyzing drought from the point of view of different disciplines revealed that each drought index rather provided a different dimension of the same drought event, which might or might not match the information obtained from other indices.. Trees in

the study region are sensitive to water deficits but their response might be highly variable depending on species. Incorporation of tree-ring information therefore resulted in years of drought in this catalogue that might lag the hydrometeorological events in time. Incorporating information on a range of drought impacts from documents provided historical context and identified certain years that were more severe due to post-war vulnerability than only a meteorological index may have suggested. The different groups of drought information provide a novel, unique set of data on drought events in southwestern Germany for

the past 218 years.

Data availability: Erfurt, M., Skiadaresis, G., Tijdeman, E., Blauhut, V., Bauhus, J., Glaser, R., Schwarz, J., Tegel, W., and Stahl, K.: Drought Catalogue Dataset, Universitätsbibliothek Freiburg/FreiDok plus, doi: 10.6094/UNIFR/166056, 2020.

Supplement: The supplement related to this article is available online at: doi inserted by NHESS

**Author contribution**: ME, GS, ET, VB and KS designed the study. ME and GS performed the analysis and prepared the manuscript with contributions from ET and VB. KS provided guidance and methodology suggestions throughout the process. All the authors read, reviewed, and approved the paper.

**Competing interests:** The authors declare that they have no conflict of interest.

**Acknowledgements**

We thank the team of R. Glaser for the provision of original text quotations of the collaborative research environment for climate and environmental history tambora.org. We acknowledge also the access to the European Drought Impact Report

Inventory database, which has initially been funded by the EU FP7 project DROUGHT-R&SPI. We would also like to express our sincere gratitude to researchers that have contributed to the international tree-ring database ITRDB (www.ncdc.noaa.gov). We acknowledge Historical Instrumental Climatological Surface Time Series of the Greater Alpine Region (HISTALP, http://www.zamg.ac.at/histalp/) for providing us with long-term meteorological data. We further thank the following agencies of the German federal states for supplying meteorological data (German Weather Service, DWD, www.dwd.de) and

streamflow data (Bundesanstalt für Gewässerkunde, BfG, www.bafg.de; Wasserstraßen- und Schifffahrtsverwaltung des

Bundes (WSV) provided by the Bundesanstalt für Gewässerkunde (BfG) & Gewässerkundlicher Dienst Bayern). Last, we thank the two anonymous referees for providing valuable suggestions and comments for improving the manuscript.

**Financial support**

This research was carried out within the interdisciplinary research project DRIeR. The project is supported by the Wassernetzwerk Baden-Württemberg (Water Research Network), which is funded by the Ministerium für Wissenschaft, Forschung und Kunst Baden-Württemberg (Ministry of Science, Research and the Arts of the State Baden-Wuerttemberg). The article processing charge was funded by the German Research Foundation (DFG) and the University of Freiburg in the funding program Open Access Publishing.

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
