# Peer review of "A multidisciplinary drought catalogue for southwestern Germany dating back to 1801"

_Natural Hazards and Earth System Sciences, 2019_

## Referee Comment (RC1) · Anonymous Referee #1 · 21 Mar 2020

**Review for manuscript "Exploring the added value of a long-term multidisciplinary dataset in drought research – a drought catalogue for southwestern Germany dating back to 1801"**

**Authors:** Mathilde Erfurt, Georgios Skiadaresis, Erik Tijdeman, Veit Blauhut, Jürgen Bauhus, Rüdiger Glaser, Julia Schwarz, Willy Tegel, and Kerstin Stahl
**Journal:** Natural Hazards and Earth System Sciences

**Summary**

Erfurt et al. present a long-term drought catalogue for southwestern Germany for the period 1801 – 2018 collected using four types of datasets: precipitation and discharge time series, tree-ring datasets, and drought impact information. They identify meteorological, hydrological, vegetation, and impact drought events using standardized time series of the four variables. They show that not all variables indicate the same events and that there are overall three periods of events with clustered drought occurrence including 1857-1870, 1947-1964, and 2003-2018. While the most severe events are visible through most variables, certain important events would not be detected if looking at just one specific time series. The study nicely highlights that the cluster of extreme events observed in the last few decades are historically not unprecedented.

**General comments**

The study compiles a long-term drought dataset considering different types of droughts. It therefore addresses two important problems in hydrology: (1) the limited record length usually available for trend analyses in extremes and (2) the multifaceted nature of drought events, which propagate through the hydrological cycle. The study is generally well written and organized even though it may profit from rephrasing and slight restructuring now and then. The datasets chosen are suitable for the analysis and the methods chosen mostly appropriate. The results are presented in clear and nicely designed figures. I below point out the need for strengthening the research question, discuss a few methodological points that should in my opinion be improved, point out a few passages that need clarification, and provide some suggestions of how the conclusions could in my view be strengthened.

**Specific comments**

1. Why not shorten the title to 'A multivariate drought catalogue for southwestern Germany dating back to 1801'? Would be a bit easier to read.
2. I think that the manuscript needs a clear research question (see e.g. abstract, where it could be added in l.16). Currently, the aim is to present a long-term drought collection. This is a methodological goal, which is fair enough. However, the paper could go further than that by asking: 'is the clustering of extreme events during the past decade unprecedented in a historical context?' I personally would frame the introduction in a way that highlights the need of a long-term dataset to answer this question. This would provide motivation for the study and highlight the practical relevance and value of the long-term dataset. The results presented allow for answering this question and lead to the conclusion that the past decade experienced frequent extreme events which is, however, not historically unprecedented if looking back into the 19[th] century.

3.  The introduction would profit from a clearer structure. I would first talk about the hazard component and the different drought types. In this first part, I would also shortly mention different drought indices (indices such as SPI but also duration, deficit, intensity, see e.g. [*Van Loon and Laaha*, 2015; *Brunner et al.*, 2019]). In a second part, I would transition to the vulnerability and impact component. Then, one could highlight the necessity of long term records to determine the rarity of certain events or periods of events. This would nicely transition to the aim of the study of providing a long-term dataset. And I would definitely talk about the value of long-term datasets in the context of trend analyses.

4.  Could you please provide a short overview of the homogenization procedure for precipitation and temperature data (l. 79)?

5.  What is the temporal resolution of the tree-ring series (l. 102)?

6.  The description of the impact dataset is a bit confusing and needs clarification (l. 105-118). Do you mean to say: 'Dataset 4 is based on reported textual information on the impacts of drought events contained in two databases'? What do you mean by 'additional reports recently collected (l. 116)? Would it be possible to provide a reference here? Could you provide a bit more information on the reasoning behind the choice of the three impact categories agriculture, ecology, and hydrological systems? Where do e.g. hydropower production and industrial water use belong to?

7.  Could you please pay attention to a consistent use of the terms 'variable', 'characteristic', 'index',… while revising the manuscript? In l. 125 e.g. do you really mean to talk about 'variables' or rather 'indices'? Or line 127: weren't indices computed from time series of anomalies?

8.  Drought definition section (l. 124-168): It remains unclear to me how exactly the drought events were determined based on the time series of indices (meteorological droughts) and percentiles (hydrological droughts). Currently, I see two aspects discussed: computation of index time series, and classification of years. Is it correct that the classification step corresponds to a threshold approach, in your case with three different thresholds? If so, could this be clarified?

9.  Computation of SPI and SPEI: why did you not use hydrological years for the computation of the index time series (l. 138)? This would be more consistent with a hydrological perspective than the use of calendar years.

10. Choice of distribution functions for derivation of SPI and SPEI: please provide a reason for the specific choices made or a suitable reference (l. 140 and 145).

11. The vegetation drought section needs some additional explanation (l. 151-161) for non-dendrochronologists: provide a reference to the 'standard methods' (l. 151), explain what a 50% frequency cutoff is (l. 154), explain what a bi-weight robust mean is (l.156), explain what an expressed population signal is (l.159).

12. Drought severity classification scheme (l. 169-178): In my understanding, this corresponds to the actual drought identification step. Could you please clarify this?

13. I think that the term 'frequencies of indices' (l. 176, l.240, l.251) is confusing (applies throughout the manuscript. If I understand this correctly, this is not a frequency but rather number of indices that co-detect a certain event. This whole part on the moving window is a bit unclear (l. 175-178). Why is this moving window approach even necessary?

14. Choice of Pearson correlation for correlation analysis (l. 186). Why use a linear correlation measure and not just a monotonic one, e.g. Kendall's or Spearman's rank correlation coefficient. Maybe there is a relationship which is just not linear.

15. 'Similarity index' (l.186-187): It remains unclear to me what exactly this index does, and why it is called similarity index. Is the ratio you are talking about n_extreme/n_all? If so, did you compute this ratio for both indices and then compare the ratios to determine similarity? Please clarify.

16. It would be nice to compute the similarity measures r and s not only for two periods but using a moving window approach allowing for an actual trend analysis (l.189-192). The problem with the two-period as opposed to a moving window approach is that one may compare a period located at the high end of an oscillation with one at the low end of an oscillation and therefore mistakenly interpret a trend even though these two periods are just located in two different parts of a cycle.

17. I do not understand why this second grouping is necessary (l. 196-200). Do you mean that you assign one or several reasons to the choice of an event?

18. No actual trend analysis is performed in this study. I would therefore not talk about 'become more frequent' (l.209) but rather say that extreme droughts happened in clusters (e.g. 1860s and recent decade). Similarly I would say 'the last decade shows a high (not higher) severity of events' (219).

19. Event clustering (l.240-249): I think that this temporal clustering aspect as opposed to a trend is interesting and deserves some more attention.

20. Figure 3: Following the methods description, would it not be more logical to present the impact panel after panel b)? Why does panel a) not have a grey background for 'no events'? In the calculation of the percentages presented in panel c, aren't the meteorological indices getting much more weight than the other indices because there is so many of them?

21. Drought frequency (section 3.2): I do not see the added value of this moving window approach. What does it allow to demonstrate which is not already shown in Figure 3c? Wouldn't some temporal clustering approach be more beneficial here? E.g. group all events separated by less than 2 years without a drought?

22. I would include Figure S5 in the main article and remove Figure 4 instead. What is the difference in the results derived from the correlation and similarity analysis? If both transport the same message, why not remove one of them?

23. Section 4.1: It is interesting to note that the droughts identified by all indices seem to have a regional extent as illustrated by the references provided. I think it would be interesting to discuss this aspect a bit further.

24. I do not think that the statement 'the recent period was characterized by higher frequency of extreme droughts' (l. 435) is particularly well supported by the results. The results presented in Figure 3 rather show that there are temporal clusters of extreme events and that the cluster of extreme events observed in the recent decade is not unprecedented (e.g. 1855-1870). I think that the strength of this study is exactly that it provides this context which is often missing when looking at short records (last 30-40) which bring us to conclusions such as 'extreme events become more frequent'. Your dataset nicely shows that periods of frequent extremes happen now but also happened in the past. I would add a discussion point on this temporal clustering aspect. Ideally, referring to existing literature.

25. I think that the conclusions could be much stronger than the ones currently presented (l.442-456). I suggest to add something along the lines of: 'Our long-term dataset shows that (1) extreme droughts cluster in time, (2) the recent decade experienced many extreme droughts similar to a period in the mid 19th century, (3) the last decade is less exceptional in a historical context than when looking at the last 30-40 years as often done in trend analyses.

26. Could you provide some information on how the community can access the dataset?

**Minor points**

l. 26: with 'potentially' widespread negative consequences …

l. 44: which 'can' affect all components …

l.70: drought types such as …

l.71: drought indices such as …

Figure 1: I would slightly extend the caption and provide a bit more information on the content of the figure. The equation within the blue box on streamflow percentiles is strange. I would remove it from the figure. Indexvalue should be 'index value'. Variable names should be in italic (e.g. T should be $T$).

l.75: 'the study employs and assembly…'. Rephrasing needed.

Figure 2: I would use a gray scale for the relief as you are just displaying one variables which does not require the use of a rainbow color scheme. By 'Stand', do you refer to 'Standort'? If yes, I would use an English abbreviation instead such as 'Loc'.

l.138: can you also provide a reason for June as you did with the other periods, analogous to the tree growth example for September?

l. 140: The distribution is fitted to the data and not the data to the distribution. Sentence needs rephrasing.

l. 142-143: reference period for what? I do not understand the meaning of this sentence.

l. 149: provide a reference for the Weibull plotting position: [*Weibull*, 1938]

l.181: do you mean from 1801-1900?

l. 184: two instead of 'three' metrics? I only see the Pearson correlation and the 'similarity index'.

l. 196: by extreme, do you refer to set D3?

l. 197: which datasets were grouped?

l. 208: I see a few more years in Figure 3b: 1964, 1949, 1991.

l. 222: Link this ET statement to literature on changes in temperature.

l. 271-271: with most, do you mean more than 10 (see caption Figure 5)? And what does the sentence 'in all cases more than 25%...) mean?

l. 288: which 'two' drought types?

Figure 5: Would it be possible to make the red color a bit more purplish to better fit into the overall color scheme?

l.304: When talking about the two different accumulation periods, do you refer to meteorological droughts?

l. 377: and due to more frequent reporting?

l. 405: how was this increase in reporting taken into account?

l.424f: use of word 'distinct', do you mean 'index-specific'. To me, the term distinct looks odd in this context.

l. 433: specify which two periods.

l.460: 'all versions of the paper'. The readers just see one.

**References used in this review**

Brunner, M. I., K. Liechti, and M. Zappa (2019), Extremeness of recent drought events in Switzerland: Dependence on variable and return period choice, *Nat. Hazards Earth Syst. Sci.*, *19*(10), 2311–2323, doi:10.5194/nhess-19-2311-2019.

Van Loon, A. F., and G. Laaha (2015), Hydrological drought severity explained by climate and catchment characteristics, *J. Hydrol.*, *526*, 3–14, doi:10.1016/j.jhydrol.2014.10.059.

Weibull, W. (1938), *Investigations into strength properties of brittle materials*, Generalstabens litografiska anstalts förlag, Stockholm.

---

## Referee Comment (RC2) · Anonymous Referee #2 · 9 Apr 2020

**Review: Erfurt et al. Exploring the added value of a long-term multidisciplinary dataset in drought research – a drought catalogue for southwestern Germany dating back to 1801**

**Summary**

This paper uses multiple datasets, including drought impacts, meteorological, hydrological and vegetation drought indices to identify droughts over ~220 years in Baden-Wuerttemberg (Germany). Drought impacts were assembled from the EDII and tambura.org and were categorised into agricultural, ecological and hydrological impacts. These were analysed alongside SPI, SPEI, tree ring data and flow percentiles from the Rhine and Danube to look at when droughts were identified (at three levels of severity) in each dataset. The events identified in each dataset were then combined to assess when droughts occurred in all datasets, and which indices identified droughts when others did not using a similarity index. The authors also compared the top 10 events identified between two time periods from 1900 and 1801, and found that there were many top 10 events in the early period between 1801 and 1900.

I believe this is an interesting, well written paper that provides a novel addition to the literature in the assessment and identification of droughts particularly due to the use of impact data. However, I do feel that the discussion could be strengthened and have noted a number of minor/more technical points that should be addressed before publication. But that these recommendations are minor, and therefore the paper can be published following minor revisions.

**Decision: Accept with minor revisions**

**Major points**

Although the title of the paper sets out the aim to explore the benefits of such a multidisciplinary approach for drought research, however I feel like the discussion would really benefit from a discussion of how these results and type of analysis is beneficial for real world applications or drought managers. In the UK for example, water companies must plan for droughts that are the worst on record (and actually now, worse than those on record using stochastic approaches), so having a good understanding of what droughts were severe, where and from what perspective is extremely important for the drought planning process. I am interested to see how this research will benefit water/drought management from the German perspective. The discussion could also be strengthened in terms out how the results compare to other studies in the region and how the droughts identified may differ in terms of their severity/impact – what was the effect of using multiple indices and the impacts here compared to these other studies?

**Minor and technical points**

L14: …Many studies have identified past drought events…

L43: It might be nice to add another one/two examples of the types of individual indices used here, as well as tree-ring based ones

L45: it felt like there was a word missing here, suggestion: …Different drought types characterised using a variety of indices…

Section 2.1: It might be nicer to use un-numbered sub-headings for each of the datasets here to make it easier for the reader to refer back to the section of interest

Figure 1: Overall I like this diagram, however have a few comments. 1) in the combined drought frequency index box you mention S1, S2, S3 events, but it's not clear from the rest of the diagram where this categorisation has come from (and I don't think is mentioned elsewhere in the paper) is there a typo here? If not, perhaps clarify? 2) The red arrows on the right of the diagram show the outcome of the meteorological, hydrological and vegetation drought indices feeding into the distinctiveness analysis, but the impact data also feed in to this section – amend the arrows to show how the impacts feed into the final part of the analysis.

L78: precipitation  totals

L78-79: I think it makes sense for the station names to be earlier in the sentence, like this '…for two stations in Baden-Wuerttemberg (Rheinstetten-Karlsruhe and Stuttgart, Figure 2), which provide the longest continuous time-series of the required variables.'

L79: It's not clear whether the 'required variables' mentioned in the first sentence are the same as the precipitation and temperature data mentioned in the second sentence – please clarify.

L95-96: You state that pine was also tested but not included because of their weak climate signal – do you have a reference or some analysis you can show (perhaps in the supplementary info) to support this?

L100: what is meant by 'appx.'?

L1157-119: You use three categories of impact (agriculture, ecology and hydrology), the EDII has ~14 categories, from what you've said I think that you have grouped EDII categories to create 3 groups of impacts, is this the case? Please clarify in the text.

Figure 2: It is quite difficult to see the colour of the points for the tree locations against the elevation layer – you could either make the elevation slightly transparent to make it paler or change the colour of the points (or both) to make it easier to read

L127: …US Drought Monitoring and Drought Impact Reporter… – capital R needed for Reporter

L133: … estimated using  the Thornthwaite equation (which only… [Space missing between 'equation' and opening bracket]

L129-145: This section on the use and calculation of SPI and SPEI would benefit from a discussion of the distributions selected and the potential impact this choice may have had on the results – e.g. recent papers have tested appropriate distributions for such standardised indices Stagge et al. 2015 (https://doi.org/10.1002/joc.4267) and Svenssson et al. 2017 (https://doi.org/10.1002/2016WR019276). It isn't clear if a reference period was used, or whether data were standardised against the whole time series available – please clarify in the text.

L151: You mention that tree ring data were gathered from 70 locations, but from looking at Figure 2 there doesn't look to be 70 points representing tree stands – is this correct (or are there many overlapping points on the map?)

L174-176: This point isn't very clear (also mentioned in the comments for Figure 1) – perhaps worth adding a section to the diagram? If the diagram gets too big for the width of the page perhaps you could add it horizontally?

L154: remove space between 50 and % symbol

L189: Both similarity measures, $r$ and $s$, were…. (r and s in italics)

L196-203: sorry this section is a bit unclear, particularly the last sentence of this paragraph. Do you mean that if one of the impact groups (e.g. tree rings) is removed, you identify 12 fewer events as these weren't identified in any of the other series? Please clarify.

L237-238: The order of the words in this sentence isn't quite right, suggestion: "For both the Rhine and the Danube, streamflow in the years 2003 and 2018 were marked as extremely low."

L240: What is the significance of the bold text here?

Figure 3: I wonder if part b might be better grouped so that all the SPI indices are together, all the SPEI indices together, all Rhine and all Danube together, with a small break or line between each group. Please add a title (as in 3a) or a yaxis label to 3c.

L270-276: It's not clear how you arrived at 17 years here from Figure 3c – please clarify. Perhaps you could mark on Figure 3c which years meet your criteria. (I think the criteria is more clearly explained in the caption for Figure 5).

L271: remove space between 25 and % symbol

Figure 5: Please explain what the numbers within each section of the rings refers to in the caption.

L302-303: This sentence doesn't read right, suggestion: The similarity index for each pair of datasets and the two periods showed some interesting patterns in the extreme droughts identified.

Figure 6 & Figure S7: group datasets/indices as suggested for Figure 3 (and all other relevant figures in Supplement). There are two grey colours on the plot, the caption indicates that grey = no extreme drought, but it doesn't say which grey this refers to. There's also some white cells in the plot – what does this mean? Please add to the legend to include white, pale grey and dark grey. The colour scale doesn't show much differentiation between colours, I recommend you add a colour to the scale (e.g. yellow) so it is easier to visually see the difference in relationship.

Figure 7: the caption is a bit unclear, I think you need to make it clear that each row shows the droughts that are identified when the group of indices are excluded from the analysis and therefore show the events unique to this group indices

L355: what's meant by double or triple drought years?

L386-388: can you comment on the changing anthropogenic influences in these catchments over time?

L417: ...ten drought events since 1900 and  since 1801 emphasised…

L420: you state here that you looked at negative impacts only, but I don't think you mention this in the data/methods section – it would be good to point this out, but also perhaps mentioning that there can be positive impacts of drought (e.g. on particular crops like strawberries)

L427: ''based on these indicators  1950…

L428: could this also be a result of improved impact data in more recent times as well?

L449-450: …Trees in the study region are  sensitive to water deficiencies …

L496: Update reference to accepted version of the paper

Figures S1-S4: perhaps these could be added as a single 4 panel plot (i.e. 2x2 grid)

Figure S5: the numbers in the cells are a bit large and in some cases merge together – they would be made a bit clearer if the text was slightly smaller (for example, it is a better size in Fig S6). Also some of the colours are quite hard to read e.g. on the TIR Fir row (perhaps this is only an issue in the low quality review figure?) Please also add a label to the colour ramp legend to say what it is showing.

Figure S6: Please also add a label to the colour ramp legend to say what it is showing. Also some of the colours are quite hard to read e.g. on the TIR Fir row (perhaps this is only an issue in the low quality review figure?)

---

## Author Comment (AC1) · 9 Apr 2020

**Reply to Referee 1**

We would like to thank reviewer 1 for his/her constructive comments and feedback on this manuscript. We think that the suggested revisions based on the reviewers' comments will improve the article. Please find our responses (in blue) to each point raised by the reviewer (shown in black) below.

**Specific comments**

1.  Why not shorten the title to 'A multivariate drought catalogue for southwestern Germany dating back to 1801'? Would be a bit easier to read.
    Response: Thank you for the suggestion, we will change the title to the version you suggested.

2.  I think that the manuscript needs a clear research question (see e.g. abstract, where it could be added in l.16). Currently, the aim is to present a long-term drought collection. This is a methodological goal, which is fair enough. However, the paper could go further than that by asking: 'is the clustering of extreme events during the past decade unprecedented in a historical context?' I personally would frame the introduction in a way that highlights the need of a long-term dataset to answer this question. This would provide motivation for the study and highlight the practical relevance and value of the long-term dataset. The results presented allow for answering this question and lead to the conclusion that the past decade experienced frequent extreme events which is, however, not historically unprecedented if looking back into the 19[th] century.
    Response: We think your suggestion of highlighting this thematic research question about the current drought cluster compared to the cluster in the past fits perfectly to our study. We will therefore emphasize this point in the next version of the manuscript.
    Most of the Authors of this paper work in the same interdisciplinary research project about droughts in Southwestern Germany (DRIeR). So, our first motivation was to collect all the different datasets we work with in our interdisciplinary team and create a common database where we can compare different drought events.

3.  The introduction would profit from a clearer structure. I would first talk about the hazard component and the different drought types. In this first part, I would also shortly mention different drought indices (indices such as SPI but also duration, deficit, intensity, see e.g. [Van Loon and Laaha, 2015; Brunner et al., 2019]). In a second part, I would transition to the vulnerability and impact component. Then, one could highlight the necessity of long term records to determine the rarity of certain events or periods of events. This would nicely transition to the aim of the study of providing a long-term dataset. And I would definitely talk about the value of long-term datasets in the context of trend analyses.
    Response: In the introduction, we aimed to highlight the need of multidisciplinary and long-term research on drought. We will apply the suggested change to first write about the hazard component, then the vulnerability and impact component, then the necessity of long-term records, this will help to further clarify our aim of this study.

4.  Could you please provide a short overview of the homogenization procedure for precipitation and temperature data (l. 79)?
    Response: The homogenisation procedure was conducted by HISTALP. We will add this information in the revised manuscript with the link to the study of Auer et al. (2007), where this homogenisation procedure is described in detail.

5. What is the temporal resolution of the tree-ring series (l. 102)?
   Response: Tree-ring series are annually resolved. Please see line 93 in the manuscript.

6. The description of the impact dataset is a bit confusing and needs clarification (l. 105-118). Do you mean to say: 'Dataset 4 is based on reported textual information on the impacts of drought events contained in two databases'? What do you mean by 'additional reports recently collected (l. 116)? Would it be possible to provide a reference here? Could you provide a bit more information on the reasoning behind the choice of the three impact categories agriculture, ecology, and hydrological systems? Where do e.g. hydropower production and industrial water use belong to?
   Response: Yes, Database 4 is based on reported textual information on the impacts of drought events contained in the European Drought Impact Report Inventory and tambora.org. We used some additional sources, which are not yet included in the database tambora.org. We will add the reference of the sources to the description. The textual information was coded after the categorisation system of the EDII.
   The categorisation scheme of the EDII is elaborate and requires detailed information on the impact, which was not always available for impacts dating further back in time. Furthermore, with respect to historic drought events, certain impacts (such as tourism, ecology) did not play a major role in daily life, and hence have not been reported as such. Accordingly, we adapted the categorisation schemes to the "major" areas of drought hazards and concern: soil moisture, hydrology and ecology (including forestry) to pool comparable impact reports. We will include a table in the Appendix, where the specific impact categories and impact types used in this study, such as hydropower production and industrial water use, are listed.

7. Could you please pay attention to a consistent use of the terms 'variable', 'characteristic', 'index',… while revising the manuscript? In l. 125 e.g. do you really mean to talk about 'variables' or rather 'indices'? Or line 127: weren't indices computed from time series of anomalies?
   Response: We will make sure that we follow the terminology in Figure 1. Indices were computed based on the variables.

8. Drought definition section (l. 124-168): It remains unclear to me how exactly the drought events were determined based on the time series of indices (meteorological droughts) and percentiles (hydrological droughts). Currently, I see two aspects discussed: computation of index time series, and classification of years. Is it correct that the classification step corresponds to a threshold approach, in your case with three different thresholds? If so, could this be clarified?
   Response: We will emphasize that we determined drought years and not drought events with a fixed start and end date. Drought years were derived from the anomaly time series of the different indices ($SPI_n$, $SPEI_n$, $Q_P$, $TRI_{species}$). A year was defined to be in drought whenever the variable of interest was abnormally low, in this study below the 20th percentile. Drought years were further classified according to three different severity classifications: D1 (moderate; <20th percentile), D2 (severe; <10th percentile) and D3 (extreme <5th percentile). We will clarify how we defined drought years in section 2.2.

9. Computation of SPI and SPEI: why did you not use hydrological years for the computation of the index time series (l. 138)? This would be more consistent with a hydrological perspective than the use of calendar years.

Response: Thank you for the question. As you mentioned in comment 20, we have already included quite some meteorological drought indices, and did not want to give more weight to meteorological drought. Deriving meteorological drought indices for the hydrological year makes sense from a hydrological perspective, but probably not for other perspectives (e.g., tree-rings are time-stamped based on the calendar year). In fact, in hydrology some countries use a 'low flow year', starting in April for routine analyses (CH) - this is close to some of our accumulation periods. Further, our aim was to show the differences between the different perspectives and not to find the best link between the different drought types.

10. Choice of distribution functions for derivation of SPI and SPEI: please provide a reason for the specific choices made or a suitable reference (l. 140 and 145).
    Response: For the SPI we selected the gamma distribution, because it best fits precipitation sums of different accumulation periods for Europe (Stagge et al., 2015). We will clarify this and add a reference in the revised manuscript. For the SPEI we used the generalized logistic distribution as suggested by Beguería et al. (2014). We will add the reference to the sentence.

11. The vegetation drought section needs some additional explanation (l. 151-161) for non-dendrochronologists: provide a reference to the 'standard methods' (l. 151), explain what a 50% frequency cutoff is (l. 154), explain what a bi-weight robust mean is (l.156), explain what an expressed population signal is (l.159).
    Response: We will provide additional information on the dendrochronological methods and statistics used.

12. Drought severity classification scheme (l. 169-178): In my understanding, this corresponds to the actual drought identification step. Could you please clarify this?
    Response: We will clarify how we defined drought years in section 2.2: Please see our response to comment 8.

13. I think that the term 'frequencies of indices' (l. 176, l.240, l.251) is confusing (applies throughout the manuscript. If I understand this correctly, this is not a frequency but rather number of indices that co-detect a certain event. This whole part on the moving window is a bit unclear (l. 175-178). Why is this moving window approach even necessary?
    Response: With our frequency analysis we want to analyse how many D1, D2 and D3 Droughts occured in a specific unit of time. For this analysis, we were not interested in the different drought types (we assume that every perspective on drought is equally important) but instead we were interested in whether drought occurrence clusters in time, i.e., whether there were decadal hotspots of increased drought occurrences. By using a moving window instead of fixed decadal time blocks, we assure that we don't miss decadal drought hot-spots that happen at the end of one decade and the beginning of the following. With your comment in mind, we see that we have to clarify that step in the revised version of the manuscript.

14. Choice of Pearson correlation for correlation analysis (l. 186). Why use a linear correlation measure and not just a monotonic one, e.g. Kendall's or Spearman's rank correlation coefficient. Maybe there is a relationship which is just not linear.
    Response: We will now repeat the analysis using Spearman's Rank-Order Correlation and compare the results obtained from the two analyses.

15. 'Similarity index' (l.186-187): It remains unclear to me what exactly this index does, and why it is called similarity index. Is the ratio you are talking about n_extreme/n_all? If so, did you compute this ratio for both indices and then compare the ratios to determine similarity? Please clarify.

Response: We will clarify in the revised manuscript that the similarity index is calculated as the total number of extreme drought years identified by two considered indices divided by the number of extreme drought years identified by index separately

$$Similarity = \frac{total\ number\ of\ extreme\ droughts/2}{number\ of\ common\ extreme\ droughts}$$

It is used to describe the similarity between two indices to identify extreme drought events.

16. It would be nice to compute the similarity measures r and s not only for two periods but using a moving window approach allowing for an actual trend analysis (l.189-192). The problem with the two-period as opposed to a moving window approach is that one may compare a period located at the high end of an oscillation with one at the low end of an oscillation and therefore mistakenly interpret a trend even though these two periods are just located in two different parts of a cycle.

Response: We agree that the selection of two periods might seem arbitrary and a moving window approach could be more appropriate here. The problem with a moving window approach, however, would be to demonstrate these results for all indices (153 combinations) and the long period examined here. We will try this approach with a reduced number of indices.

17. I do not understand why this second grouping is necessary (l. 196-200). Do you mean that you assign one or several reasons to the choice of an event?

Response: Figure 5 and 7 grouping was necessary to identify similarities and differences among the datasets. Additionally we want to distinguish between short- and long-term meteorological drought indices (SPI and SPEI). There was almost no difference between the long-term SPEIs and the long-term SPIs, therefore we excluded the latter one for Fig. 5 and 7. We will add more information on the grouping in order to clarify that point.

18. No actual trend analysis is performed in this study. I would therefore not talk about 'become more frequent' (l.209) but rather say that extreme droughts happened in clusters (e.g. 1860s and recent decade). Similarly I would say 'the last decade shows a high (not higher) severity of events' (219).

Response: Indeed we did not perform any trend analysis, so we changed the sentence to "several clusters of increased drought occurrence were identified". With the sentence "the last decade shows a higher severity of drought events" we mean, that more drought events are classified as extreme (class D3). We will change the sentence accordingly.

19. Event clustering (l.240-249): I think that this temporal clustering aspect as opposed to a trend is interesting and deserves some more attention.

Response: We will emphasize more on the temporal clustering in the revised manuscript.

20. Figure 3: Following the methods description, would it not be more logical to present the impact panel after panel b)? Why does panel a) not have a grey background for 'no events'? In the calculation of the percentages presented in panel c, aren't the meteorological indices getting much more weight than the other indices because there is so many of them?

Response: We decided to present the catalogue in this order, because the impacts were not included in panel c). Based on your comment, we see that this seems to be confusing, so we will change the order of the plot. In the revised manuscript we will present Figure 3 in the following

order: (a) individual indices, (b) composite information, (c) impacts.
We decided against a grey background for "no events" in the impact dataset, because unlike the other datasets used in this study, we cannot guarantee that we have evaluated all available written sources. That means, we cannot say for sure if there was "no event". We will expand the point on challenges with textual information in the discussion. In panel c) the meteorological indices make up a large proportion of the number of events per year, but this proportion does not change through time, so the 'weight' stays the same.  The percentage of indices indicating droughts was calculated based on the number of available indices per year, i.e. adjusted for the period following 1900 when Q became available.

21. Drought frequency (section 3.2): I do not see the added value of this moving window approach. What does it allow to demonstrate which is not already shown in Figure 3c? Wouldn't some temporal clustering approach be more beneficial here? E.g. group all events separated by less than 2 years without a drought?
Response: Please see also our response to comment 13. In Fig. 3c we only show the number of drought events per severity class per year. In the next step, we were interested in the temporal clustering of droughts, i.e., decadal drought hot spots. Therefore we decided to calculate how many droughts per severity class occurred in 5 and 15-year periods. Instead of the moving windows, we could have used fixed timespans (e.g. for 5 years starting in 1801: 1801 to 1806, 1807 to 1811 and so on) but this is then biased by the starting point, so we found it more objective to use the moving window approach.

22. I would include Figure S5 in the main article and remove Figure 4 instead. What is the difference in the results derived from the correlation and similarity analysis? If both transport the same message, why not remove one of them?
Response: Do you mean Figure 6 instead of Figure 4? In the similarity analysis we used only extreme droughts (severity class D3), while in the correlation analysis (Figure S5) we included all severity classes. We will include the formula we provided in our response to point 15, we hope this will demonstrate the differences between the two metrics.

23. Section 4.1: It is interesting to note that the droughts identified by all indices seem to have a regional extent as illustrated by the references provided. I think it would be interesting to discuss this aspect a bit further.
Response: Indeed, that is an interesting point which we will include in the discussion of the revised manuscript.

24. I do not think that the statement 'the recent period was characterized by higher frequency of extreme droughts' (l. 435) is particularly well supported by the results. The results presented in Figure 3 rather show that there are temporal clusters of extreme events and that the cluster of extreme events observed in the recent decade is not unprecedented (e.g. 1855- 1870). I think that the strength of this study is exactly that it provides this context which is often missing when looking at short records (last 30-40) which bring us to conclusions such as 'extreme events become more frequent'. Your dataset nicely shows that periods of frequent extremes happen now but also happened in the past. I would add a discussion point on this temporal clustering aspect. Ideally, referring to existing literature.
Response: We agree with you. We will change this sentence to: "was characterized by increased occurrence of extreme droughts." Nevertheless, within our dataset especially in the last five years

(see Fig. 4c, last bar: 2014 to 2018) we found more extreme droughts (severity class D3) than in the other timespans. But we agree with you, that the clustering of several years with extreme droughts is not unprecedented. Both points are important, so we will distinguish between them and focus also on the temporal clustering in the discussion.

25. I think that the conclusions could be much stronger than the ones currently presented (l.442- 456). I suggest to add something along the lines of: 'Our long-term dataset shows that (1) extreme droughts cluster in time, (2) the recent decade experienced many extreme droughts similar to a period in the mid 19$^{th}$ century, (3) the last decade is less exceptional in a historical context than when looking at the last 30-40 years as often done in trend analyses.

Response: Thank you very much for your suggestions. We will include these points in the conclusion along with the value of using a multivariate dataset. We think this will nicely underline the main aspects of this study: I) using a long-term and II) and multivariate dataset for regional drought research.

26. Could you provide some information on how the community can access the dataset?

Response: Yes, we are currently looking for an option to provide access to the dataset used here.

**Minor points**

Thank you for the minor comments. We agree with the comments and suggestion of small changes and will address all minor points in the revised manuscript.

**References used in this review and the reply to this review**

Auer, I., Böhm, R., Jurkovic, A., Lipa, W., Orlik, A., Potzmann, R., Schöner, W., Ungersböck, M., Matulla, C., Briffa, K. R., Jones, P., Efthymiadis, D., Brunetti, M., Nanni, T., Maugeri, M., Mercalli, L., Mestre, O., Moisselin, J.-M., Begert, M., Müller-Westermeier, G., Kveton, V., Bochnicek, O., Stastny, P., Lapin, M., Szalai, S., Szentimrey, T., Cegnar, T., Dolinar, M., Gajic-Capka, M., Zaninovic, K., Majstorovic, Z., and Nieplova, E.: HISTALP - historical instrumental climatological surface time series of the Greater Alpine Region, International Journal of Climatology, 27, 17–46, https://doi.org/10.1002/joc.1377, 2007.

Beguería, S., Vicente-Serrano, S. M., Reig, F., and Latorre, B.: Standardized precipitation evapotranspiration index (SPEI) revisited: Parameter fitting, evapotranspiration models, tools, datasets and drought monitoring, Int. J. Climatol., 34, 3001–3023, https://doi.org/10.1002/joc.3887, 2014.

Brunner, M. I., K. Liechti, and M. Zappa (2019), Extremeness of recent drought events in Switzerland: Dependence on variable and return period choice, *Nat. Hazards Earth Syst. Sci.*, *19*(10), 2311– 2323, doi:10.5194/nhess-19-2311-2019.

Stagge, J. H., Tallaksen, L. M., Gudmundsson, L., van Loon, A. F., and Stahl, K.: Candidate Distributions for Climatological Drought Indices (SPI and SPEI), Int. J. Climatol., 35, 4027–4040, https://doi.org/10.1002/joc.4267, 2015.

Van Loon, A. F., and G. Laaha (2015), Hydrological drought severity explained by climate and catchment characteristics, *J. Hydrol.*, *526*, 3–14, doi:10.1016/j.jhydrol.2014.10.059.

Weibull, W. (1938), *Investigations into strength properties of brittle materials*, Generalstabens litografiska anstalts förlag, Stockholm.

---

## Author Comment (AC2) · 20 Apr 2020

**Reply to Referee 2**

We would like to thank Referee 2 for his/her constructive comments and feedback on this manuscript. We think that the suggested revisions based on the Referee's comments will certainly improve the article. Please find our responses (in blue) to each point raised by Referee 2 (shown in black) below.

**Major points**

Although the title of the paper sets out the aim to explore the benefits of such a multidisciplinary approach for drought research, however I feel like the discussion would really benefit from a discussion of how these results and type of analysis is beneficial for real world applications or drought managers. In the UK for example, water companies must plan for droughts that are the worst on record (and actually now, worse than those on record using stochastic approaches), so having a good understanding of what droughts were severe, where and from what perspective is extremely important for the drought planning process. I am interested to see how this research will benefit water/drought management from the German perspective. The discussion could also be strengthened in terms out how the results compare to other studies in the region and how the droughts identified may differ in terms of their severity/impact – what was the effect of using multiple indices and the impacts here compared to these other studies?

Response: 1) We like the idea of discussing the benefits or the potential of our study for drought management and will include it in section 4.2. This is indeed crucial given the lack of a centralized drought management authority in southwestern Germany. For drought monitoring the State Institute for the Environment of Baden-Württemberg might offer a low flow monitoring system in near future and the DWD provides monthly maps of an aridity index for Germany but there is no overall drought management authority in place. With our drought catalogue we get a better understanding of past drought events. The worst case droughts can be used as inputs for future drought planning. We suggest to provide a discussion point where we stress the need of a multi-sectoral regional drought management plan and how this or similar multidisciplinary studies could provide a basis and help to develop such a plan. 2) In section 4.2 we will also include a comparison to other drought studies in our region to discuss the added value of a multivariate drought study instead of a single variable drought study (even though most drought studies in the study region focus on methodological aspects).

**Minor and technical points**

Thank you for the minor comments. We agree with your suggestions and comments on grammatical changes and minor technical points and will address them in the revised manuscript. Please find below our replies to some comments that we thought needed some clarification or discussion.

Figure 1: Overall I like this diagram, however have a few comments. 1) in the combined drought frequency index box you mention S1, S2, S3 events, but it's not clear from the rest of the diagram where this categorisation has come from (and I don't think is mentioned elsewhere in the paper) is there a typo here? If not, perhaps clarify? 2) The red arrows on the right of the diagram show the outcome of the meteorological, hydrological and vegetation drought indices feeding into the

distinctiveness analysis, but the impact data also feed in to this section – amend the arrows to show how the impacts feed into the final part of the analysis.

Response: to 1) Indeed, that was a mistake. We will change that to D1, D2 and D3 for the different drought classes. 2) Thanks for the hint! We will add the missing arrow.

L1157-119: You use three categories of impact (agriculture, ecology and hydrology), the EDII has ~14 categories, from what you've said I think that you have grouped EDII categories to create 3 groups of impacts, is this the case? Please clarify in the text.

Response: Yes, we have grouped the EDII categories into three new groups of impacts (agriculture, ecology and hydrology). We will include a Table in the supplement where we list which impact type from the EDII categorisation was assigned to the three groups and will also clarify this in the text.

Figure 2: It is quite difficult to see the colour of the points for the tree locations against the elevation layer – you could either make the elevation slightly transparent to make it paler or change the colour of the points (or both) to make it easier to read

Response: We will change this Figure and use less colors.

L129-145: This section on the use and calculation of SPI and SPEI would benefit from a discussion of the distributions selected and the potential impact this choice may have had on the results – e.g. recent papers have tested appropriate distributions for such standardised indices Stagge et al. 2015 (https://doi.org/10.1002/joc.4267) and Svensssson et al. 2017 (https://doi.org/10.1002/2016WR019276). It isn't clear if a reference period was used, or whether data were standardised against the whole time series available – please clarify in the text.

Response: We will clarify that we selected the gamma distribution for the SPI, because of its general good fit with precipitation sums of different accumulation periods for Europe and refer to Stagge et al., 2015. For the SPEI, we clarify that we used the generalized logistic distribution as suggested by Beguería et al. (2014, https://doi.org/10.1002/joc.3887) and refer to the latter paper. As reference period we used the longest available common period for both meteorological stations (which is 1810 to 2018). We will clarify this.

L151: You mention that tree ring data were gathered from 70 locations, but from looking at Figure 2 there doesn't look to be 70 points representing tree stands – is this correct (or are there many overlapping points on the map?)

Response: Indeed, that was a mistake. The initial dataset consisted of tree-ring data from 70 locations. After quality control and removal of short series the final dataset consisted of tree-ring series from 55 locations (see also Table S1). We will change this in L151 and use open symbols in Figure 2 to improve visibility of overlapping locations.

L174-176: This point isn't very clear (also mentioned in the comments for Figure 1) – perhaps worth adding a section to the diagram? If the diagram gets too big for the width of the page perhaps you could add it horizontally?

Response: We agree that this sentence needs clarification. However, it is already included in Figure 1 (Drought Severity Classification). In this part we described that a year was defined to be in drought whenever the variable of interest was abnormally low, in this study below the 20th percentile. Drought years were further classified according to three different severity classifications: D1 (moderate; <20th percentile), D2 (severe; <10th percentile) and D3 (extreme <5th percentile). We will clarify how we

L196-203: sorry this section is a bit unclear, particularly the last sentence of this paragraph. Do you mean that if one of the impact groups (e.g. tree rings) is removed, you identify 12 fewer events as these weren't identified in any of the other series? Please clarify.

Response: Yes, in this analysis we assess which extreme droughts we would miss if we exclude one of the groups of indices. We will clarify this in the revised version and change the last sentence of the paragraph to "The distinctiveness value is the number of extreme droughts we would miss if we exclude a specific dataset."

Figure 3: I wonder if part b might be better grouped so that all the SPI indices are together, all the SPEI indices together, all Rhine and all Danube together, with a small break or line between each group. Please add a title (as in 3a) or a yaxis label to 3c.

Response: We think in this figure it is more important to have the accumulation periods together because we want to emphasize the difference between SPI and SPEI (which is more clearly visible by plotting the SPI-3 below the SPEI-3) and not the differences between accumulation periods. We will add the label to Figure 3c as suggested.

L270-276: It's not clear how you arrived at 17 years here from Figure 3c – please clarify. Perhaps you could mark on Figure 3c which years meet your criteria. (I think the criteria is more clearly explained in the caption for Figure 5).

Response: In the revised manuscript we will include in L. 270, that we focused on the 20 drought years, where most indices point to a drought. In L. 272 we will rearrange the sentence to "After 1900 all 17 years (the period for which streamflow data was available) were identified as drought years based on streamflow indices."

Figure 5: Please explain what the numbers within each section of the rings refers to in the caption.

Response: They refer to the number of indices per group pointing to a drought. We will clarify this in the revised manuscript.

Figure 6 & Figure S7: group datasets/indices as suggested for Figure 3 (and all other relevant figures in Supplement). There are two grey colours on the plot, the caption indicates that grey = no extreme drought, but it doesn't say which grey this refers to. There's also some white cells in the plot – what does this mean? Please add to the legend to include white, pale grey and dark grey. The colour scale doesn't show much differentiation between colours, I recommend you add a colour to the scale (e.g. yellow) so it is easier to visually see the difference in relationship.

Response: In Figure 6 and S7 we used only one grey color. The grey color used just seems darker when surrounded by dark red boxes. The white cells indicate 0 similarity while the grey boxes indicate that no extreme drought was identified by either of the two indices. We will clarify this in the legend and figure caption. In the revised version of the manuscript we will use symbols instead of grey color to avoid confusion.

L355: what's meant by double or triple drought years?

Response: Here we meant 'two or three consecutive years of extreme droughts'. We will clarify the sentence.

L386-388: can you comment on the changing anthropogenic influences in these catchments over time?

Response: We will add a sentence on this in the discussion. In general, human influences on river flow are susceptible to changes over time and further provide some detail and references how (changing) anthropogenic influences might have affected flow in the considered rivers.

L428: could this also be a result of improved impact data in more recent times as well?

Response: We agree, the lack of distinct events in the last decades might be the result of both improved impact and hydrometeorological data. We will add a comment on this.

---

## Author Response (AR1)

**List of relevant changes**

We would like to thank both Referees for their valuable comments and suggestions on this manuscript.

We largely followed their comments and recommendations. Relevant changes made are:

- The restructured and rewritten introduction,
- The better explanation of the drought severity classification scheme (section 2.2.5),
- The now separated sections 4.2 and 4.3 and the new section 4.4 in the discussion

Please find our responses (in green) to each point raised by the Referees (shown in black) and a marked-up manuscript version below.

**List of Changes for Referee 1**

We would like to thank Referee 1 for his/her constructive comments and feedback on this manuscript. We think that the suggested revisions based on the reviewers' comments helped improve the article. Please find our responses (in green) to each point raised by the reviewer (shown in black) below.

**General comments**

The study compiles a long-term drought dataset considering different types of droughts. It therefore addresses two important problems in hydrology: (1) the limited record length usually available for trend analyses in extremes and (2) the multifaceted nature of drought events, which propagate through the hydrological cycle. The study is generally well written and organized even though it may profit from rephrasing and slight restructuring now and then. The datasets chosen are suitable for the analysis and the methods chosen mostly appropriate. The results are presented in clear and nicely designed figures. I below point out the need for strengthening the research question, discuss a few methodological points that should in my opinion be improved, point out a few passages that need clarification, and provide some suggestions of how the conclusions could in my view be strengthened.

Response: We followed your suggestions and rephrased, restructured and focused the points you specify below.

**Specific comments**

1. Why not shorten the title to 'A multivariate drought catalogue for southwestern Germany dating back to 1801'? Would be a bit easier to read.
   Response: Thank you for the suggestion, we changed the title to the version you suggested (but with using the term "multidisciplinary").

2. I think that the manuscript needs a clear research question (see e.g. abstract, where it could be added in l.16). Currently, the aim is to present a long-term drought collection. This is a methodological goal, which is fair enough. However, the paper could go further than that by asking: 'is the clustering of extreme events during the past decade unprecedented in a historical context?' I personally would frame the introduction in a way that highlights the need of a long-term dataset to answer this question. This would provide motivation for the study and highlight the practical relevance and value of the long-term dataset. The results presented allow for answering this question and lead to the conclusion that the past decade experienced frequent extreme events which is, however, not historically unprecedented if looking back into the 19[th] century.
   Response: We think your suggestion of highlighting this thematic research question about the current drought cluster compared to the cluster in the past fits perfectly to our study. We therefore emphasized this point in the new version of the manuscript. Throughout the manuscript we now strengthened the need of long-term data in drought research. We included a new purpose point c) in the introduction (l. 79). We deepened the point in the result part of the manuscript (see section 3.1 and 3.2) and we emphasized the importance of this point in a new discussion section 4.2.

3. The introduction would profit from a clearer structure. I would first talk about the hazard component and the different drought types. In this first part, I would also shortly mention

different drought indices (indices such as SPI but also duration, deficit, intensity, see e.g. [*Van Loon and Laaha*, 2015; *Brunner et al.*, 2019]). In a second part, I would transition to the vulnerability and impact component. Then, one could highlight the necessity of long term records to determine the rarity of certain events or periods of events. This would nicely transition to the aim of the study of providing a long-term dataset. And I would definitely talk about the value of long-term datasets in the context of trend analyses.

Response: In the introduction, we aimed to highlight the need of multidisciplinary and long-term research on drought. We applied the suggested change to first write about the hazard component and indices, then the vulnerability and impact component, then the necessity of long-term records to further clarify our aim of this study.

4. Could you please provide a short overview of the homogenization procedure for precipitation and temperature data (l. 79)?

   Response: The homogenisation procedure was conducted by HISTALP. We added this information in the revised manuscript with the link to the study of Auer et al. (2007), where the homogenisation procedure is described in detail (l.102).

5. What is the temporal resolution of the tree-ring series (l. 102)?

   Response: Thank you for the comment, tree-ring series are annually resolved (please see l. 117).

6. The description of the impact dataset is a bit confusing and needs clarification (l. 105-118). Do you mean to say: 'Dataset 4 is based on reported textual information on the impacts of drought events contained in two databases'? What do you mean by 'additional reports recently collected (l. 116)? Would it be possible to provide a reference here? Could you provide a bit more information on the reasoning behind the choice of the three impact categories agriculture, ecology, and hydrological systems? Where do e.g. hydropower production and industrial water use belong to?

   Response: Yes, Database 4 is based on reported textual information on the impacts of drought events contained in the European Drought Impact Report Inventory and tambora.org. We used some additional sources (e.g. Alzenauer Wetterchronik), which are not yet included in the database tambora.org. We added the reference of the sources to the description (l. 139-140). The textual information was coded after the categorisation system of the EDII.
   The categorisation scheme of the EDII is elaborate and requires detailed information on the impact, which was not always available for impacts dating further back in time. Furthermore, with respect to historic drought events, certain impacts (such as tourism, ecology) did not play a major role in daily life, and hence have not been reported as such. Accordingly, we adapted the categorisation schemes to the "major" areas of drought hazards and concern: soil moisture, hydrology and ecology (including forestry) to pool comparable impact reports. We included a Table S2 in the Supplement material, where the specific impact categories and impact types used in this study, such as hydropower production and industrial water use, are listed.

7. Could you please pay attention to a consistent use of the terms 'variable', 'characteristic', 'index',… while revising the manuscript? In l. 125 e.g. do you really mean to talk about 'variables' or rather 'indices'? Or line 127: weren't indices computed from time series of anomalies?

   Response: We follow now the terminology in Figure 1. Indices were computed based on the

variables. In l. 125 we really meant variables (like precipitation data) that were used to calculate indices. In l. 127 we meant indices and corrected that in the revised manuscript (l. 153 now).

8. Drought definition section (l. 124-168): It remains unclear to me how exactly the drought events were determined based on the time series of indices (meteorological droughts) and percentiles (hydrological droughts). Currently, I see two aspects discussed: computation of index time series, and classification of years. Is it correct that the classification step corresponds to a threshold approach, in your case with three different thresholds? If so, could this be clarified?
Response: We now emphasize that we determined drought years and not drought events with a fixed start and end date. Drought years were derived from the anomaly time series of the different indices ($SPI_n$, $SPEI_n$, $Q_P$, $TRI_{species}$). A year was defined to be in drought whenever the variable of interest was abnormally low, in this study below the 20th percentile. Drought years were further classified according to three different severity classifications: D1 (moderate; <20th percentile), D2 (severe; <10th percentile) and D3 (extreme <5th percentile). We have now clarified how we defined drought years in section 2.2.

9. Computation of SPI and SPEI: why did you not use hydrological years for the computation of the index time series (l. 138)? This would be more consistent with a hydrological perspective than the use of calendar years.
Response: Thank you for the question. As you mentioned in comment 20, we have already included quite many meteorological drought indices, and did not want to give more weight to meteorological drought. Deriving meteorological drought indices for the hydrological year makes sense from a hydrological perspective, but probably not for other perspectives (e.g., tree-rings are time-stamped based on the calendar year). In fact, in hydrology some countries use a 'low flow year', starting in April for routine analyses (e.g. CH) - this is close to some of our accumulation periods. Further, our aim was to show the differences between the different perspectives and not to find the best link between the different drought types.

10. Choice of distribution functions for derivation of SPI and SPEI: please provide a reason for the specific choices made or a suitable reference (l. 140 and 145).
Response: For the SPI we selected the gamma distribution, because it best fits precipitation sums of different accumulation periods for Europe (Stagge et al., 2015). We clarified this and added a reference in the revised manuscript. For the SPEI we used the generalized logistic distribution as suggested by Beguería et al. (2014). We added the reference to the sentence (l. 169.173).

11. The vegetation drought section needs some additional explanation (l. 151-161) for non-dendrochronologists: provide a reference to the 'standard methods' (l. 151), explain what a 50% frequency cutoff is (l. 154), explain what a bi-weight robust mean is (l.156), explain what an expressed population signal is (l.159).
Response: We have now provided additional information on the dendrochronological methods and statistics used (l. 190-195).

12. Drought severity classification scheme (l. 169-178): In my understanding, this corresponds to the actual drought identification step. Could you please clarify this?
Response: We clarified how we defined drought years in section 2.2: "Drought years were

derived from the anomaly time series of the different indices (SPIn, SPEIn, QP, TRIspecies). A year was defined to be in drought whenever the variable of interest was abnormally low, in this study below the 20th percentile. Drought years were further classified according to three different severity classifications: D1 (moderate; <20th percentile), D2 (severe; <10th percentile) and D3 (extreme <5th percentile)." Please see our response to comment 8.

13. I think that the term 'frequencies of indices' (l. 176, l.240, l.251) is confusing (applies throughout the manuscript. If I understand this correctly, this is not a frequency but rather number of indices that co-detect a certain event. This whole part on the moving window is a bit unclear (l. 175-178). Why is this moving window approach even necessary?
Response: With our "frequency" analysis we want to analyse how many D1, D2 and D3 Droughts occured in a specific unit of time. We see that the word "frequency" is not quite clear in that context. We therefore changed it throughout the manuscript to the word "combined drought index" and "occurrence" of droughts co-detect by several indices.
For this analysis, we were not interested in the different drought types (we assume that but every perspective on drought is equally important) but instead we were interested in whether drought occurrence clusters in time, i.e., whether there were decadal hotspots of increased drought occurrences. By using a moving window instead of fixed decadal time blocks, we assure that we don't miss decadal drought hot-spots that happen at the end of one decade and the beginning of the following. We clarified that step in section 2.2 (l.214-219), in the figure description of Fig. 3, and in section 3.2.

14. Choice of Pearson correlation for correlation analysis (l. 186). Why use a linear correlation measure and not just a monotonic one, e.g. Kendall's or Spearman's rank correlation coefficient. Maybe there is a relationship which is just not linear.
Response: Thank you for this comment. We have now repeated the analysis using Spearman's Rank-Order correlation (see figure below) and compared the results obtained from the two analyses. The correlation coefficients obtained from the two analyses are almost identical.

[Figure]

15. 'Similarity index' (l.186-187): It remains unclear to me what exactly this index does, and why it is called similarity index. Is the ratio you are talking about n_extreme/n_all? If so, did you compute this ratio for both indices and then compare the ratios to determine similarity? Please clarify.
Response: We will clarify in the revised manuscript that the similarity index is calculated as the total number of extreme drought years identified by two considered indices divided by the number of extreme drought years identified by index separately:

$$Similarity = \frac{total\ number\ of\ extreme\ droughts/2}{number\ of\ common\ extreme\ droughts}$$

It is used to describe the similarity between two indices to identify extreme drought events. We clarified this in the revised manuscript version and added the formula in line 232 to better explain our steps.

16. It would be nice to compute the similarity measures r and s not only for two periods but using a moving window approach allowing for an actual trend analysis (l.189-192). The problem with the two-period as opposed to a moving window approach is that one may compare a period located at the high end of an oscillation with one at the low end of an oscillation and therefore mistakenly interpret a trend even though these two periods are just located in two different parts of a cycle.
Response: We agree that the selection of two periods might seem arbitrary and a moving

window approach could be more appropriate here. The problem with a moving window approach, however, would be to demonstrate these results for all indices (153 combinations) and the long period examined here.

17. I do not understand why this second grouping is necessary (l. 196-200). Do you mean that you assign one or several reasons to the choice of an event?

    Response: In Figures 5 and 7 grouping was necessary to identify similarities and differences among the datasets. Additionally we want to distinguish between short- and long-term meteorological drought indices (SPI and SPEI). There was almost no difference between the long-term SPEIs and the long-term SPIs, therefore we excluded the latter one for Fig. 5 and 7. We added more information on the grouping in order to clarify that point (line 243-245).

18. No actual trend analysis is performed in this study. I would therefore not talk about 'become more frequent' (l.209) but rather say that extreme droughts happened in clusters (e.g. 1860s and recent decade). Similarly I would say 'the last decade shows a high (not higher) severity of events' (219).

    Response: Indeed we did not perform any trend analysis, so we changed the sentence to "several clusters of increased drought occurrence were identified" (line 252-254). With the sentence "the last decade shows a higher severity of drought events" we mean, that more drought events are classified as extreme (class D3). We changed the sentence accordingly (line 260-261).

19. Event clustering (l.240-249): I think that this temporal clustering aspect as opposed to a trend is interesting and deserves some more attention.

    Response: We emphasized more on the temporal clustering in the revised manuscript in section 3.1 and 3.2.

20. Figure 3: Following the methods description, would it not be more logical to present the impact panel after panel b)? Why does panel a) not have a grey background for 'no events'? In the calculation of the percentages presented in panel c, aren't the meteorological indices getting much more weight than the other indices because there is so many of them?

    Response: We decided to present the catalogue in this order, because the impacts were not included in panel c). Based on your comment, we see that this seems to be confusing, so we will change the order of the plot. In the revised manuscript we will present Figure 3 in the following order: (a) individual indices, (b) impacts, (c) composite information. We expanded the point on challenges with textual information in the discussion (line 444-447). In panel c) the meteorological indices make up a large proportion of the number of events per year, but this promotion does not change through time, so the 'weight' stays the same. The percentage of indices indicating droughts was calculated based on the number of available indices per year. This was corrected for the period following 1900 when Q was available.

21. Drought frequency (section 3.2): I do not see the added value of this moving window approach. What does it allow to demonstrate which is not already shown in Figure 3c? Wouldn't some temporal clustering approach be more beneficial here? E.g. group all events separated by less than 2 years without a drought?

    Response: Please see also our response to comment 13. In Fig. 3c we only show the number of

drought events per severity class per year. In the next step, we were interested in the temporal clustering of droughts, i.e., decadal drought hot spots. Therefore we decided to calculate how many droughts per severity class occurred in 5 and 15-year periods. Instead of the moving windows, we could have used fixed timespans (e.g. for 5 years starting in 1801: 1801 to 1806, 1807 to 1811 and so on) but this is then biased by the starting point, so we found it more objective to use the moving window approach.

22. I would include Figure S5 in the main article and remove Figure 4 instead. What is the difference in the results derived from the correlation and similarity analysis? If both transport the same message, why not remove one of them?
Response: Do you mean Figure 6 instead of Figure 4? In the similarity analysis we used only extreme droughts (severity class D3), while in the correlation analysis (Figure S5) we included all severity classes. We now include the formula we provided in our response to point 15, we hope this now demonstrates the differences between the two metrics.

23. Section 4.1: It is interesting to note that the droughts identified by all indices seem to have a regional extent as illustrated by the references provided. I think it would be interesting to discuss this aspect a bit further.
Response: Indeed, that is an interesting point which we extended slightly in the revised manuscript (section 4.1).

24. I do not think that the statement 'the recent period was characterized by higher frequency of extreme droughts' (l. 435) is particularly well supported by the results. The results presented in Figure 3 rather show that there are temporal clusters of extreme events and that the cluster of extreme events observed in the recent decade is not unprecedented (e.g. 1855- 1870). I think that the strength of this study is exactly that it provides this context which is often missing when looking at short records (last 30-40) which bring us to conclusions such as 'extreme events become more frequent'. Your dataset nicely shows that periods of frequent extremes happen now but also happened in the past. I would add a discussion point on this temporal clustering aspect. Ideally, referring to existing literature.
Response: We agree and instead now wrote "was characterized by increased occurrence of extreme droughts." Nevertheless, within our dataset especially the last five years (see Fig. 4c, last bar: 2014 to 2018) we find more extreme droughts (severity class D3) then in the other timespans. But we agree with you, that the clustering of several years with extreme droughts is not unprecedented. Both points are important, so we distinguish between them and focus also on the temporal clustering in the discussion (see new Section 4.2).

25. I think that the conclusions could be much stronger than the ones currently presented (l.442-456). I suggest to add something along the lines of: 'Our long-term dataset shows that (1) extreme droughts cluster in time, (2) the recent decade experienced many extreme droughts similar to a period in the mid 19th century, (3) the last decade is less exceptional in a historical context than when looking at the last 30-40 years as often done in trend analyses.
Response: We included these points in the conclusion beside the points of using a multivariate dataset. We think this now underlines the main aspects of this study: I) using a long-term and II) multivariate dataset for regional drought research.

26. Could you provide some information on how the community can access the dataset?
    Response: Yes, we will provide access to the drought catalogue data. The doi is reserved and data will be uploaded upon acceptance of the paper.

**Minor points**

l. 26: with 'potentially' widespread negative consequences ...
Response: We changed the sentence as suggested (l. 27).

l. 44: which 'can' affect all components ...
Response: We changed the sentence (l. 28).

l.70: drought types such as ...
Response: We added the drought types we referred to (l. 85).

l.71: drought indices such as ...
Response: We added some examples (l. 87).

Figure 1: I would slightly extend the caption and provide a bit more information on the content of the figure. The equation within the blue box on streamflow percentiles is strange. I would remove it from the figure. Indexvalue should be 'index value'. Variable names should be in italic (e.g. T should be $T$).
Response: Thank you for the comment. We extended the caption. We removed the equation on streamflow percentiles. We also changed 'Indexvalues' to 'Index values' and made variable names italic.

l.75: 'the study employs and assembly...'. Rephrasing needed.
Response: We rephrased the sentence (l. 90).

Figure 2: I would use a gray scale for the relief as you are just displaying one variables which does not require the use of a rainbow color scheme. By 'Stand', do you refer to 'Standort'? If yes, I would use an English abbreviation instead such as 'Loc'.
Response: The term stand is used to refer to the location of forest stands (https://en.wikipedia.org/wiki/Forest_stand). We changed the figure and use a grey background now.

l.138: can you also provide a reason for June as you did with the other periods, analogous to the tree growth example for September?
Response: We now provide a reason for June as well (l. 164).

l. 140: The distribution is fitted to the data and not the data to the distribution. Sentence needs rephrasing.
Response: We rephrased the sentence (l. 166-167).

l. 142-143: reference period for what? I do not understand the meaning of this sentence.
Response: We rephrased the sentence (l. 169).

l. 149: provide a reference for the Weibull plotting position: [Weibull, 1938]
Response: We added the reference (l. 178).

l.181: do you mean from 1801-1900?

Response: We changed the sentence to "we ranked the ten most extreme years since 1900 onward for all indicators, and additionally since 1801 onward for the meteorological and the tree-ring dataset" and we hope it is now clear that we mean 1900-2017/18 and 1800-2017/18.

l. 184: two instead of 'three' metrics? I only see the Pearson correlation and the 'similarity index'.

Response: Here we refer to the three metrics (correlation, similarity and distinctiveness) used to quantify similarities and relationships among the different drought indices. We changed the text accordingly (l.225-229).

l. 196: by extreme, do you refer to set D3?

Response: Yes, we added D3 to the sentence in the revised manuscript (l. 239).

l. 197: which datasets were grouped?

Response: We meant 'indices'. We changed that in the revised manuscript (l. 240).

l. 208: I see a few more years in Figure 3b: 1964, 1949, 1991.

Response: Yes, there are more years. We listed only some examples. Nevertheless, we now include more examples (l. 252).

l. 222: Link this ET statement to literature on changes in temperature.

Response: We changed the sentence and included a reference to the ET statement in the discussion section 4.2 (l. 463).

l. 271-271: with most, do you mean more than 10 (see caption Figure 5)? And what does the sentence 'in all cases more than 25%...) mean?

Response: Yes, with 'most' we meant 'more than' 10. With 'more than 25%' we refer to the dotted line in Fig. 3c). We changed the sentence in the new version (l. 300 and the caption of Figure 5).

l. 288: which 'two' drought types?

Response: We rephrased the sentence to "meteorological and hydrological drought indicators" (l. 338).

Figure 5: Would it be possible to make the red color a bit more purplish to better fit into the overall color scheme?

Response: See new version of Figure 5.

l.304: When talking about the two different accumulation periods, do you refer to meteorological droughts?

Response: No, here we refer to the two average periods we used for stream flow (Q.March-November and Q.June-November). We clarified this in the text (l. 352-353).

l. 377: and due to more frequent reporting?

Response: Yes, the increased awareness (private, public and governmental) might be expressed by more frequent reporting (e.g. by an increase in reports on ecological impacts (Blauhut et al 2015). We clarified that in the new version of the manuscript (l. 446).

l. 405: how was this increase in reporting taken into account?

Response: We used binary information (annual value indicating 'impact' or 'no impact' occurrence in each category). We corrected the increase in reporting by converting every year before 1947 with one or more indicated impact into an impact year with IDI=1. In the period 1947-1999, years with more than two reported impacts were characterized as impact years. After 2000, years with more than three reported

impacts were considered as impact years (see section 2.2.4).

l.424f: use of word 'distinct', do you mean 'index-specific'. To me, the term distinct looks odd in this context.
Response: We changed the word to 'unique'.

l. 433: specify which two periods.
Response: We specified the two periods (l. 451).

l.460: 'all versions of the paper'. The readers just see one.
Response: Yes, that's true. We changed it to 'the paper'.

**References used in this review and the reply to this review**

Auer, I., Böhm, R., Jurkovic, A., Lipa, W., Orlik, A., Potzmann, R., Schöner, W., Ungersböck, M., Matulla, C., Briffa, K. R., Jones, P., Efthymiadis, D., Brunetti, M., Nanni, T., Maugeri, M., Mercalli, L., Mestre, O., Moisselin, J.-M., Begert, M., Müller-Westermeier, G., Kveton, V., Bochnicek, O., Stastny, P., Lapin, M., Szalai, S., Szentimrey, T., Cegnar, T., Dolinar, M., Gajic-Capka, M., Zaninovic, K., Majstorovic, Z., and Nieplova, E.: HISTALP - historical instrumental climatological surface time series of the Greater Alpine Region, International Journal of Climatology, 27, 17–46, https://doi.org/10.1002/joc.1377, 2007.

Beguería, S., Vicente-Serrano, S. M., Reig, F., and Latorre, B.: Standardized precipitation evapotranspiration index (SPEI) revisited: Parameter fitting, evapotranspiration models, tools, datasets and drought monitoring, Int. J. Climatol., 34, 3001–3023, https://doi.org/10.1002/joc.3887, 2014.

Brunner, M. I., K. Liechti, and M. Zappa (2019), Extremeness of recent drought events in Switzerland: Dependence on variable and return period choice, *Nat. Hazards Earth Syst. Sci.*, *19*(10), 2311– 2323, doi:10.5194/nhess-19-2311-2019.

Stagge, J. H., Tallaksen, L. M., Gudmundsson, L., van Loon, A. F., and Stahl, K.: Candidate Distributions for Climatological Drought Indices (SPI and SPEI), Int. J. Climatol., 35, 4027–4040, https://doi.org/10.1002/joc.4267, 2015.

Van Loon, A. F., and G. Laaha (2015), Hydrological drought severity explained by climate and catchment characteristics, *J. Hydrol.*, *526*, 3–14, doi:10.1016/j.jhydrol.2014.10.059.

Vicente-Serrano, S. M., Beguería, S., and López-Moreno, J. I.: A Multiscalar Drought Index Sensitive to Global Warming: The Standardized Precipitation Evapotranspiration Index, J. Climate, 23, 1696–1718, https://doi.org/10.1175/2009JCLI2909.1, 2010.

Weibull, W. (1938), *Investigations into strength properties of brittle materials*, Generalstabens litografiska anstalts förlag, Stockholm.

**List of Changes for Referee 2**

We would like to thank Referee 2 for his/her constructive comments and feedback on this manuscript. We think that the suggested revisions based on the reviewers' comments improved the article. Please find our responses (in green) to each point raised by the reviewer (shown in black) below.

**Major points**

Although the title of the paper sets out the aim to explore the benefits of such a multidisciplinary approach for drought research, however I feel like the discussion would really benefit from a discussion of how these results and type of analysis is beneficial for real world applications or drought managers. In the UK for example, water companies must plan for droughts that are the worst on record (and actually now, worse than those on record using stochastic approaches), so having a good understanding of what droughts were severe, where and from what perspective is extremely important for the drought planning process. I am interested to see how this research will benefit water/drought management from the German perspective. The discussion could also be strengthened in terms out how the results compare to other studies in the region and how the droughts identified may differ in terms of their severity/impact – what was the effect of using multiple indices and the impacts here compared to these other studies?

Response: We like the idea of discussing the benefits or the potential of our study for drought management and included a new section 4.4. This is indeed crucial given the lack of a centralized drought management authority in southwestern Germany. Drought monitoring is dispersed:  state level river situation monitoring existits and there are two federal level platforms by UfZ focusing on soil moisture anomalies for agricultural land and the DWD provides monthly maps of an aridity index for Germany but there is no overall, multi-sectoral  drought management authority in place. With our drought catalogue we get a better understanding of past drought events. The worst case droughts can be used as benchmark events  to inform  future drought planning. We provide a discussion point 4.4 where we stress the need of a multi-sectoral regional drought management plan and how this or similar multidisciplinary studies could provide a basis and help to develop such a plan.

Further, we now focus in the discussion in section  4.1 on the data used and compare the identified droughts to other drought studies in Europe (most drought studies in the study region focus on methodological aspects and are only based on single variables). We discuss now separately the added value of long-term datasets in section 4.2, and the value of multidisciplinary studies in drought research in section 4.3.

**Minor and technical points**

L14: …Many studies have identified past drought events…

Response: We changed the sentence accordingly (L15).

L43: It might be nice to add another one/two examples of the types of individual indices used here, as well as tree-ring based ones

Response: We added some more examples to this part (paragraph beginning L56).

L45: it felt like there was a word missing here, suggestion: …Different drought types characterised using a variety of indices…
Response: We changed the whole paragraph (L40).

Section 2.1: It might be nicer to use un-numbered sub-headings for each of the datasets here to make it easier for the reader to refer back to the section of interest
Response: We inserted a third level of sub-headings for each dataset. We did the same in section 2.2.

Figure 1: Overall I like this diagram, however have a few comments. 1) in the combined drought frequency index box you mention S1, S2, S3 events, but it's not clear from the rest of the diagram where this categorisation has come from (and I don't think is mentioned elsewhere in the paper) is there a typo here? If not, perhaps clarify? 2) The red arrows on the right of the diagram show the outcome of the meteorological, hydrological and vegetation drought indices feeding into the distinctiveness analysis, but the impact data also feed in to this section – amend the arrows to show how the impacts feed into the final part of the analysis.
Response: 1) Indeed, that was a mistake. We changed the description to D1, D2 and D3 or the different drought classes. 2) We added the missing arrow.

L78: precipitation  totals
Response: In literature we find both words for precipitation sums/totals. We decided to keep the word "sums".

L78-79: I think it makes sense for the station names to be earlier in the sentence, like this '…for two stations in Baden-Wuerttemberg (Rheinstetten-Karlsruhe and Stuttgart, Figure 2), which provide the longest continuous time-series of the required variables.'
Response: We change the sentence as suggested (L99).

L79: It's not clear whether the 'required variables' mentioned in the first sentence are the same as the precipitation and temperature data mentioned in the second sentence – please clarify.
Response: Yes, they are the same. We clarified this (L100).

L95-96: You state that pine was also tested but not included because of their weak climate signal – do you have a reference or some analysis you can show (perhaps in the supplementary info) to support this?
Response: Thank you for this comment. Indeed, this was not shown anywhere in our results and we believe this information is not necessary. We have, therefore, deleted this sentence in the revised version of the manuscript.

L100: what is meant by 'appx.'?
Response: We meant the appendix. But now we refer to the supplementary material with Table S1 and so on and therefore delete 'appx'.

L1157-119: You use three categories of impact (agriculture, ecology and hydrology), the EDII has ~14 categories, from what you've said I think that you have grouped EDII categories to create 3 groups of impacts, is this the case? Please clarify in the text.

Response: Yes, we have grouped the EDII categories into three new groups of impacts (agriculture, ecology and hydrology). We included a Table S2 in the supplements where we list which impact type from the EDII categorisation was used for the three groups and we clarified this in the text.

Figure 2: It is quite difficult to see the colour of the points for the tree locations against the elevation layer – you could either make the elevation slightly transparent to make it paler or change the colour of the points (or both) to make it easier to read
Response: We changed the figure. Elevation layer is now excluded, we use a grey background instead.

L127: …US Drought Monitoring and Drought Impact Rreporter… – capital R needed for Reporter
Response: We changed that (L153).

L133: … estimated using with the Thornthwaite equation (which only… [Space missing between 'equation' and opening bracket]
Response: We changed the sentence accordingly (L159).

L129-145: This section on the use and calculation of SPI and SPEI would benefit from a discussion of the distributions selected and the potential impact this choice may have had on the results – e.g. recent papers have tested appropriate distributions for such standardised indices Stagge et al. 2015 (https://doi.org/10.1002/joc.4267) and Svensson et al. 2017 (https://doi.org/10.1002/2016WR019276). It isn't clear if a reference period was used, or whether data were standardised against the whole time series available – please clarify in the text.
Response: For the SPI we selected the gamma distribution, because it best fits precipitation sums of different accumulation periods for Europe and referred to Stagge et al., 2015. For the SPEI, we clarify that we used the generalized logistic distribution as suggested by Beguería et al. (2014, https://doi.org/10.1002/joc.3887 ) and refer to the latter paper. As reference period we used the longest available common period for both meteorological stations (which is 1810 to 2018). We clarified this.

L151: You mention that tree ring data were gathered from 70 locations, but from looking at Figure 2 there doesn't look to be 70 points representing tree stands – is this correct (or are there many overlapping points on the map?)
Response: That was indeed a mistake. The initial dataset consisted of tree-ring data from 70 locations. Following our selection procedure (L181) we included only tree-ring series from 55 locations. Still several points in Figure 2 are overlapping.

L174-176: This point isn't very clear (also mentioned in the comments for Figure 1) – perhaps worth adding a section to the diagram? If the diagram gets too big for the width of the page perhaps you could add it horizontally?
Response: We agree that this sentence needs clarification. However, it is already included in Fig. 1 (Drought Severity Classification). In this part we described that a year was defined to be in drought whenever the variable of interest was abnormally low, in this study below the 20th percentile. Drought years were further classified according to three different severity classifications: D1 (moderate; <20th percentile), D2 (severe; <10th percentile) and D3 (extreme <5th percentile). We clarified in the revised version how we defined drought years in section 2.2.

L154: remove space between 50 and % symbol

Response: In NHESS spaces must be included between number and unit. During the "typesetting" of NHESS, the "correct" spaces between symbols and numbers will probably be added.

L189: Both similarity measures, *r* and *s*, were…. (r and s in italics)

Response: We changed it to italics  (L227).

L196-203: sorry this section is a bit unclear, particularly the last sentence of this paragraph. Do you mean that if one of the impact groups (e.g. tree rings) is removed, you identify 12 fewer events as these weren't identified in any of the other series? Please clarify.

Response: This is exactly what the distinctiveness value stands for. We have now clarified that in the text (L240).

L237-238: The order of the words in this sentence isn't quite right, suggestion: "For both the Rhine and the Danube, streamflow in the years 2003 and 2018 were marked as extremely low."

Response: We changed the sentence accordingly  (L265).

L240: What is the significance of the bold text here?

Response: There is no significance, therefore we unbolded the text  (L283).

Figure 3: I wonder if part b might be better grouped so that all the SPI indices are together, all the SPEI indices together, all Rhine and all Danube together, with a small break or line between each group. Please add a title (as in 3a) or a yaxis label to 3c.

Response: We think in this figure it is more important to have the accumulation periods together because we want to emphasize the difference between SPI and SPEI (which is more clearly visible by plotting the SPI-3 below the SPEI-3) and not the differences between accumulation periods (see Fig. below). In the revised manuscript we present Figure 3 in the following order: (a) individual indices, (b) impacts, (c) composite information.

[Figure]

**Fig.: Drought catalogue. Indices grouped by type (SPI, SPEI, Danube and Rhine) as well as the accumulation periods.**

L270-276: It's not clear how you arrived at 17 years here from Figure 3c – please clarify. Perhaps you could mark on Figure 3c which years meet your criteria. (I think the criteria is more clearly explained in the caption for Figure 5).

Response: In the revised manuscript we included in L321, that we focused on the 20 drought years, in which most indices point to a drought. In L323 we rearranged the sentence to "After 1900 all 17 years (the period for which streamflow data was available) were identified as drought years based on streamflow indices."

L271: remove space between 25 and % symbol

Response: In NHESS spaces must be included between number and unit.

Figure 5: Please explain what the numbers within each section of the rings refers to in the caption.

Response: They refer to the number of indices per group pointing to a drought. We clarified this in the caption of Figure 5.

L302-303: This sentence doesn't read right, suggestion: The similarity index for each pair of datasets and the two periods showed some interesting patterns in the extreme droughts identified.

Response: We have now changed the sentence based on your suggestion (L351).

Figure 6 & Figure S7: group datasets/s as suggested for Figure 3 (and all other relevant figures in Supplement). There are two grey colours on the plot, the caption indicates that grey = no extreme drought, but it doesn't say which grey this refers to. There's also some white cells in the plot – what does this mean? Please add to the legend to include white, pale grey and dark grey. The colour scale doesn't show much differentiation between colours, I recommend you add a colour to the scale (e.g. yellow) so it is easier to visually see the difference in relationship.

Response: In Figure 6 and S7 we used only one grey color. The grey color used just seems darker when surrounded by dark red boxes. The white cells indicate 0 similarity while the grey boxes indicate that no extreme drought was identified by either of the two indices. We have added a black rectangle around the colour ramp which we believe clarifies this point.

Figure 7: the caption is a bit unclear, I think you need to make it clear that each row shows the droughts that are identified when the group of indices are excluded from the analysis and therefore show the events unique to this group indices

Response: We rewrote the caption of Fig. 7 and additionally use the word 'category' instead of 'group' now.

L355: what's meant by double or triple drought years?

Response: Here we meant 'two or three consecutive years of extreme drought'. We clarified the sentence (L405).

L386-388: can you comment on the changing anthropogenic influences in these catchments over time?

Response: We added a sentence on this in the discussion (L427-428). In general, human influences on river flow are susceptible to changes over time and further provide some detail and references how (changing) anthropogenic influences might have affected flow in the considered rivers.

L417: ...ten drought events since 1900 and  since 1801 emphasised...
Response: We changed the sentence as suggested (L450-451).

L420: you state here that you looked at negative impacts only, but I don't think you mention this in the data/methods section – it would be good to point this out, but also perhaps mentioning that there can be positive impacts of drought (e.g. on particular crops like strawberries)
Response: We now mention it in the data section (L129).

L427: ''based on these indicators  1950...
Response: We changed this sentence to "after 1950" (L477).

L428: could this also be a result of improved impact data in more recent times as well?
Response: We agree, the lack of distinct events in the last decades might be the result of both improved impact and hydrometeorological data. We added a comment on this (L478-479).

L449-450: ...Trees in the study region are  sensitive to water deficiencies ...
Response: We changed the sentence accordingly (L512).

L496: Update reference to accepted version of the paper
Response: We updated the reference to the accepted version of the paper (L569).

Figures S1-S4: perhaps these could be added as a single 4 panel plot (i.e. 2x2 grid)
Response: We made a single plot as suggested by you. See Fig. S1 in the supplementary material.

Figure S5: the numbers in the cells are a bit large and in some cases merge together – they would be made a bit clearer if the text was slightly smaller (for example, it is a better size in Fig S6). Also some of the colours are quite hard to read e.g. on the TIR Fir row (perhaps this is only an issue in the low quality review figure?) Please also add a label to the colour ramp legend to say what it is showing.
Response: We have decreased the size of the numbers and included a legend to the color ramp (see Fig. S2).

Figure S6: Please also add a label to the colour ramp legend to say what it is showing. Also some of the colours are quite hard to read e.g. on the TIR Fir row (perhaps this is only an issue in the low quality review figure?)
Response: We have included a legend to the colour ramp (see Fig. S3).

[revised manuscript text omitted]

---

## Referee Report (RR1)

**Review for manuscript "A multidisciplinary drought catalogue for southwestern Germany dating back to 1981"**

**Authors:** Mathilde Erfurt, Georgios Skiadaresis, Erik Tijdeman, Veit Blauhut, Jürgen Bauhus, Rüdiger Glaser, Julia Schwarz, Willy Tegel, and Kerstin Stahl
**Journal:** Natural Hazards and Earth System Sciences

**Summary**

Erfurt et al. present a long-term drought catalogue for southwestern Germany for the period 1801 – 2018 collected using four types of datasets: precipitation and discharge time series, tree-ring datasets, and drought impact information. The authors have seriously considered the two reviewers' comments, which has helped to improve the structure and methods description.

**General comments**

I thank the authors for addressing the points risen in my review of the first version of the manuscript. I think that it will be a good fit for NHESS after some language polishing. I provide some suggestions below and would recommend going through the whole manuscript again with a focus on the consistent use of tense (past vs. present), use of clear and simple sentence structures, choice of verbs, use of commas, and use of prepositions. For my taste, too many remarks are provided in between brackets. I would try to integrate them in the text directly in order to avoid interruption of the reading flow.

**Suggested edits**

l. 20: temporal patterns

l. 35: SSI seems to be a more commonly used abbreviation than SSFI

l.38-39: Can in my opinion be removed

l. 43: A single variable assessment provided sector relevant information. Analyzing the root zone soil moisture drought signal for example provides information relevant for agriculture.

l. 44: not necessarily linked in a similar way in different catchments?

l. 52-53: yes, but maybe you would like to acknowledge the impact data sources you are using in this study?

l. 55: potentially impacted instead of that may be impacted.

l. 56: Knowledge on past… (delete The)

l. 57: delete the in front of extremes and knowledge.

l. 65: 250 years derived using…

l. 69: in the countries part of the Danube?

l. 70: All these existing catalogues?

l. 85: remove the in front of meteorological

l. 86: add comma after datasets

l. 89: add comma after the bracket.

l. 165: SPI and SPEI were computed using…

l. 169-170. I think this sentence is redundant with l. 167 and can be removed.

l. 166-168: I would shortly comment on the stationarity assumption.

l. 172: I would shortly mention the gof test used for the distribution fitting and report the p-values obtained.

l. 174-178: I was wondering here why you did not use the standardized streamflow index to be consistent with the methods chosen for meteorological drought. I would add a short explanation here.

l. 186: A bi-weight robust mean, which reduces…

l. 194: SNR has been introduced above and the abbreviation can be used here in isolation.

l. 206: I would specify how many indices were considered in total and per variable.

l. 209: under drought

l. 213: events identified using their…  events identified according to…

l. 218: we assured that we did not

l. 227: for extreme events only.

l. 231: I would write average number in the equation instead of total number/2 to be consistent with l. 230.

l. 233: would remove 'both'

l. 239: remove 'the' in front of extreme droughts.

l. 249: replace in by at

l.251: several years stick out

l. 258: what does 'they' refer to?

l. 260: remove 'the' in front of potential

l. 264: remove in the streamflow dataset because this information is redundant.

l. 329: indicate instead of indicating.

l. 334: For 11 out

Figure 6: use lower caps for 'between'

l.409-411: sentence in my opinion needs rephrasing.

l. 439: Although care was taken when simplifying the impact dataset into three categories, the results…

l. 458: This highlights the need for…

l. 480: value of drought catalogue for drought management?

l. 512: I do not see the purpose of the sentence 'Time of occurrence of hydrometeorological water deficits is only one feature' as timing is not really considered in this study.

---

## Author Response (AR2)

[revised manuscript text omitted]
 reference period for parameter fitting was set to the longest available period, in this case 1810 to 2018. The SPEI was calculated in the same way, but based on the climatic water balance (difference between potential precipitation and evapotranspiration). For the SPEI, standardization was based on the generalized logistic distribution (Beguería et al., 2014).

**2.2.2 Hydrological drought**

Hydrological drought was calculated from daily streamflow observations (Dataset 2). The daily streamflow *(Q)* data for the period between 1901-2018 was aggregated to annual as well as seasonal averages for both the non-winter (March-November) and the summer and autumn (June-November) seasons. The aggregated streamflow data were then transferred to streamflow percentiles ($Q_P$), using Weibull plotting positions ($Q_P$ = rank(Q) / (n+1); where n in this case equals the amount of years) (Weibull, 1938).

**2.2.3 Vegetation drought**

Vegetation drought indexing followed standard dendrochronological methods (e.g. Speer, 2010) in order to derive a tree-ring index (TRI) from Dataset 3. The 2089 different tree-ring series originated from 55 locations in Baden-Wuerttemberg (Fig. 2 and Table S1). To remove age-related growth trends from individual tree-ring series while maintaining their inter-annual variability, we detrended raw ring-width series using a 30 year spline with 50 % frequency response cutoff (the frequency at which 50 % of the amplitude of a signal is retained, see also Cook et al., 2013). This commonly used detrending approach removes the biological trends present in growth series (low frequency) while simultaneously preserving annual to decadal variability in growth (high frequency) (Cook and Peters, 1981; Speer, 2010). A bi-weight robust mean, which (whereby reducing thereduces the influence of outliers in the computation of the mean), was then calculated to generate four residual chronologies: oak, fir, spruce and a combined chronology including trees from all three species (see Fig. S1). The quality of the developed chronologies was assessed using several descriptive statistics (EPS: expressed population signal, SNR: signal to noise ratio, and rbar: mean interseries correlation) commonly used in dendrochronology (Speer, 2010; Table S1). The EPS is an indicator of how well a chronology represents a theoretical infinite population (Wigley et al., 1984). Low values of EPS (commonly <0.85) indicate that the chronologies are dominated by individual tree signals rather than a consistent regional signal (Speer, 2010). Rbar is the mean correlation between series within a chronology and is a measure of common signal strength of detrended chronologies. The signal to noise ratio (SNR) is a measure of the desired signal in each chronology

[revised manuscript text omitted]